

# Fitting the junction model and other parameterizations for the unsaturated soil hydraulic conductivity curve: KRIAfitter version 1.0

## Gerrit H. de Rooij[1]

[1]Dept. Soil System Science, Helmholtz Centre for Environmental Research – UFZ, Halle, 06120, Germany

*Correspondence to*: Gerrit H. de Rooij (gerrit.derooij@ufz.de)

**Abstract.** Several current models for the unsaturated soil hydraulic conductivity curve consider the conductivity of the domains of capillary water in water-filled pores and adsorbed water in films on soil grains, as well as an equivalent conductivity for water vapour diffusion. These models rely on unrealistic configuration of the domains. A junction model is introduced that sidesteps this problem by assigning all liquid water to films (dry range), or to capillaries (wet range). Combined with a sigmoidal junction model for the soil water retention curve, it has up to six fitting parameters, one less than the other multidomain models. Tests on data for 13 soils show that the junction model and an additive model (that adds all domain conductivities) often produce good fits. Models with six or more parameters may be overparameterized for many soils, giving the more parsimonious junction model an advantage, but for some soils, the extra parameter of the additive model is needed to achieve a good fit. This paper and a User Manual document a Fortran code (KRIAfitter) that uses the Shuffled Complex Evolution algorithm to fit the junction, additive, and four other conductivity models for any combination of fixed and fitting parameters or their log-transforms. KRIAfitter either maps the Root Mean Square Error in the entire parameter space in order to then constrain the parameter space around the likely global minimum, or it generates many fits and uses those to calculate statistics for individual parameters, as well as the covariance and correlation matrices.

## 1 Introduction

Madi et al. (2018) revealed that many parameterizations of the soil water retention curve (SWRC) gave physically unrealistic near-saturated behaviour of the soil hydraulic conductivity when applied in combination with Kosugi's (1999) generalized soil hydraulic conductivity parameterization. This motivated de Rooij et al. (2021) and de Rooij (2022) to propose a closed-form expression for the SWRC with a distinct air-entry value, like Ippisch et al. (2006), a sigmoid shape in the intermediate range according to van Genuchten (1980), and a logarithmic dry branch terminating at a finite matric potential at which the soil was oven-dry, with the water content essentially zero. The volumetric water contents and derivatives of the sigmoid and logarithmic branches were matched at the matric





potential of their junction according to Rossi and Nimmo (1994). This SWRC model (termed RIA, for Rossi-Ippisch-
Adaptation) had a finite slope near saturation, which remedied the problems of most existing parameterizations, and
also the asymptotic residual water content at the dry end.

De Rooij et al. (2021) presented an analytical expression for a soil hydraulic conductivity curve that could be
used with this new SWRC-parameterization. This conductivity curve was a special case of Kosugi's (1999)
parameterization. The public discussion of de Rooij et al.'s paper (accessible on-line) revealed the desirability of an
alternative formulation that could be fitted separately and had the capability to account for non-capillary flow.

Peters and Durner (2008) and Peters (2013) included non-capillary flow in their model for the unsaturated
hydraulic conductivity curve (UHCC) by separating the total liquid water in a domain with adsorbed water (present as
films on the surface of the solid phase) and a domain with capillary bound water. They combined parametric models
for soil water retention and unsaturated hydraulic conductivity for both. By assuming instantaneous equilibrium
between water vapour pressure and the matric potential of liquid water, Peters (2013) could also formulate a model
for isothermal vapour flow driven by the gradient in the matric potential. Weber et al. (2019) formalized this approach
in a modular set-up facilitating different choices for the parameterizations chosen to represent the SWRC and the UHCC
of the various domains. In all three papers, the adsorbed and capillary-bound water contents were added to arrive at
the total water content. The hydraulic conductivities were also added to find the total conductivity at a given matric
potential. De Rooij (2024a) posited that this additivity attribute requires that all flow domains are arranged in parallel,
and developed averaging models using the arithmetic and harmonic means of the domain conductivities for domains
in parallel and in series, respectively. He also offered a model using the geometric mean as an intermediate between
the other two, as well as a non-weighted additive model akin to that of Peters (2013). From his analysis it appears
fundamentally impossible to derive the bulk soil hydraulic conductivity from domain conductivities based on domain
volumes and configurations.

Given these complications, this paper introduces a new model, in which the film domain and the capillary
domain do not exist in parallel, but instead are joined at a critical matric potential below which all liquid water is in
films and above which all water is capillary-bound. This creates a junction model for the soil hydraulic conductivity
analogously to the RIA parameterization for the SWRC. The vapour domain is assumed to be parallel to the liquid-water
domain. Together with this new model, the paper documents a Fortran program (KRIAfitter) that is able to fit the
parameters of the junction model and all models introduced by de Rooij (2024a) to unsaturated hydraulic
conductivities observed for either different water contents or different matric potentials (de Rooij, 2024b). All models
operate in conjunction with the RIA parameterization of de Rooij (2022). The fitting code (RIAfitter) for that model
was thoroughly overhauled for this study, which resulted in version 2.0 (de Rooij, 2024c). While KRIAfitter is the main
focus of this paper, it is expected that RIAfitter and KRIAfitter will normally be used in tandem.





## 2. The junction models for the soil water retention and hydraulic conductivity curves

### 2.1 The model for the soil water retention curve

This section presents the main equations from de Rooij (2022) for clarity. He presented a unimodal model for the SWRC.


$$\theta(h) = \begin{cases} 0, & h \le (1+c)h_d \\ \theta_s \beta \ln\left[\frac{(1+c)h_d}{h}\right], & (1+c)h_d < h \le h_{j\theta} \\ \theta_s \left(\frac{1+|\alpha h|^n}{1+|\alpha h_{ae}|^n}\right)^{\frac{1}{n}-1}, & h_{j\theta} < h \le h_{ae} \\ \theta_s, & h > h_{ae} \end{cases} \tag{1a}$$

where $h$ denotes the matric potential in equivalent water column (L), subscripts $d$ and $ae$ denote the value at which the water content reaches zero and the air-entry value, respectively, and subscript $j\theta$ indicates the value of $h$ at which the

logarithmic and sigmoid branch are joined. The volumetric water content is denoted by $\theta$, with the subscript $s$ denoting its value at saturation. Parameters $\alpha$ (L$^{-1}$) and $n$ determine the shape of the sigmoid branch (van Genuchten, 1980), while parameter $\beta$ does so for the logarithmic branch. By requiring the derivatives of the sigmoidal and logarithmic branches to match at $h_{j\theta}$, parameter $\beta$ can be expressed in terms of the other parameters (de Rooij et al., 2021).

$$\beta = (n-1)|\alpha h_{j\theta}|^n (1+|\alpha h_{ae}|^n)^{1-\frac{1}{n}} \left(1+|\alpha h_{j\theta}|^n\right)^{\frac{1}{n}-2} \tag{1b}$$

De Rooij (2022) made $h_{j\theta}$ a derived parameter with the following expression.

$$h_{j\theta} = h_d e^{\frac{1}{1-n}} \tag{1c}$$


The five fitting parameters then are: $h_{ae}$, $h_d$, $\theta_s$, $\alpha$ and $n$. Equation (1c) does not guarantee continuity at $h_{j\theta}$, so de Rooij (2022, 2024a) introduced a correction factor that is very small for most soils.

$$c = \exp\left\{\frac{1}{(n-1)\left[|\alpha h_d|\exp\left(\frac{1}{1-n}\right)\right]^n}\right\} - 1 \tag{1d}$$


This correction needs to be applied to $h_d$ in the logarithmic branch of the SWRC as shown in Eq. (1a). No correction is needed in the UHCC.



### 2.2 The junction model for the unsaturated hydraulic conductivity curve

A junction model analogous to that used for the SWRC above gives a monotonically increasing UHCC with
increasing matric potential. It requires that liquid soil water is entirely allocated to either capillary-bound water or to
water adsorbed in films, with the water abruptly changing its allocation when the matric potential passes through the
matric potential at the junction point $h_j$ (L), which will not necessarily be equal to $h_{j\theta}$. Because the liquid phase in this
model only occupies a single domain at any given matric potential, the issue of the correct averaging of the domain
conductivities with its complications (de Rooij, 2024a) becomes moot. The fact that $K_a$ and $K_c$ are intrinsic
conductivities strictly valid for their respective domains instead of bulk conductivities (de Rooij, 2024a) is resolved
implicitly through the fitted values of the parameters, made possible because averaging of domain conductivities is not
needed.

The clear distinction between adsorbed and capillary-bound water of Eqs. (1a–d) appears well suited for use
with the multi-domain conceptualization of Peters and Durner (2008) and Peters (2013), and with the modular
framework presented by Weber et al. (2019). Weber et al.'s (2019) approach is tailored to additive formulations of the
SWRC in which capillary-bound and adsorbed water co-exist over the full moisture range, but the SWRC of Eqs. (1a–d)
has no capillary-bound water for $h < h_j$. Furthermore, a water film on a soil particle is bounded on one side by a solid–
liquid interface and on the other by a liquid–gas interface. The velocity profile in the water film is such that its gradient
at the liquid–gas interface equals zero (Eq. (8) of Or and Tuller, 2000). When a pore with a water film on its grain
surfaces takes in additional water and becomes fully saturated, the region previously occupied by the water film now
only has a solid–liquid interface, while at the location of the former liquid–gas interface, there is now moving water and
a non-zero gradient in the velocity profile. This will increase the flow rate in the region previously occupied by the film.
Simple addition of film and capillary conductivities may therefore not be accurate. For these reasons, the conductivity
expressions were adapted as outlined below.

The intrinsic hydraulic conductivity of water in films is modeled according to Peters (2013).

$$K_a(h) = K_{s,a} \cdot \begin{cases} 0, & h \leq (1+c)h_d \\ \left(\frac{h}{h_a}\right)^{-1.5}, & (1+c)h_d < h \leq h_a \\ 1, & h > h_a \end{cases} \tag{2}$$

$K_{s,a}$ (LT$^{-1}$) is the value of $K_a$ when the domain with adsorbed water is completely filled, and $h_a$ (L) is the matric potential
at which this occurs. The value of the exponent is adopted from Peters (2013). Note that $K_a(h)$ abruptly drops to zero
at $(1+c)h_d$, but $K_a$ at that matric potential is so small that this will generally be insignificant for practical use.





In the junction model, $h_a$ is set equal to $h_j$. For the range in which capillary-bound water is present, Kosugi's (1999) model is used with $\kappa = 1$ (see de Rooij, 2024a) for the soil hydraulic conductivity as if all water is capillary-bound, irrespective of the matric potential.


$$K_{c,jun}(h) = K_{s,c,jun} \cdot \begin{cases} \left(\frac{G(h)}{G(h_{ae})}\right)^{\tau} \left(\frac{1-|\alpha h|^{n-1}G(h)}{1-|\alpha h_{ae}|^{n-1}G(h_{ae})}\right)^{\gamma}, & h_j < h \leq h_{ae} \\ 1, & h > h_{ae} \end{cases} \tag{3a}$$

where

$$G(x) = (1 + |\alpha x|^n)^{\frac{1}{n}-1} \tag{3b}$$

Here, $K_{c,jun}$ is the unsaturated hydraulic conductivity of the capillary domain in the junction model (LT$^{-1}$) and $K_{s,c,jun}$ its value at saturation. Mualem (1976) proposed the value 0.5 for $\tau$ and 2.0 for $\gamma$. Assouline (2001) introduced a simpler expression by setting $\tau$ to 0.0.

130        Equations (2) and (3a–b) establish a continuous conductivity curve if their values are matched at $h_j$. This requires that the following equality holds.

$$K_{s,a} = K_{s,c,jun} \left(\frac{G(h_j)}{G(h_{ae})}\right)^{\tau} \left(\frac{1-|\alpha h_j|^{n-1}G(h_j)}{1-|\alpha h_{ae}|^{n-1}G(h_{ae})}\right)^{\gamma} \tag{4}$$

135        Vapour flow is assumed to be diffusive under isothermal conditions. The equilibrium between local vapour pressure and matric potential is assumed to be instantaneous. The model used is de Rooij's (2024a) modification of that of Peters (2013). The equivalent water vapour bulk hydraulic conductivity $K_v^{\beta}$ is:

$$K_v^B(h) = \frac{\rho_{sv} M g \ a}{\rho_w R (T+273.15)\theta_s^2}(\theta_s - \theta)^{\frac{10}{3}} e^{\frac{Mg}{R(T+273.15)}h} \tag{5a}$$


Because of the specific nature of the expressions and their constants, units replace dimensions in the explanations of the variables and parameters. The units of $K_v^{\beta}$ are cm d$^{-1}$, consistent with those for the liquid water conductivity. The densities of saturated water vapour and liquid water are $\rho_{sv}$ and $\rho_w$ (kg m$^{-3}$), respectively, $M$ (kg mol$^{-1}$) is the molar mass of water, $g$ (cm d$^{-2}$) is the gravitational acceleration, $R$ is the universal gas constant (kg cm$^2$ d$^{-2}$ K$^{-1}$ mol$^{-1}$), and $T$

is the temperature (°C).



The densities $\rho_w$ and $\rho_{sv}$ are temperature-dependent. Peters (2013) provides an expression for the temperature dependence of $\rho_{sv}$:

$$\rho_{sv} = \frac{0.0010}{T}\exp\left[31.3716 - \frac{6014.79}{T} - 0.00792495T\right] \tag{5b}$$


The density of liquid water in the range between 0 and 40 °C was approximated according to Brutsaert (2005, p. 17).

$$\rho_w = 999.8505 + 0.06001T - 0.007917T^2 + 4.1256 \cdot 10^{-5}T^3 \tag{5c}$$

For the temperature-dependency of $D_a$, the expression of Dorsey (1940, p. 73) was converted to units cm and day, and adapted to a reference temperature of 15 °C.

$$D_a = 2.09 \cdot 10^4 \left(\frac{T+273.15}{288.15}\right)^{1.75} \tag{5d}$$

The bulk vapour conductivity model does not have any fitting parameters.

Combining Eqs. (2), (3a), (4), and (5a) gives the expression for the bulk hydraulic conductivity to water according to the junction model, denoted $K_{jun}^B$ (LT$^{-1}$).

$$K_{jun}^B(h) = K_v^B(h) + K_{s,c,jun} \cdot \begin{cases} 0, & h \le (1+c)h_d \\ \left(\frac{G(h_j)}{G(h_{ae})}\right)^\tau \left(\frac{1-|\alpha h_j|^{n-1}G(h_j)}{1-|\alpha h_{ae}|^{n-1}G(h_{ae})}\right)^\gamma \left(\frac{h}{h_j}\right)^{-1.5}, & (1+c)h_d < h \le h_j \\ \left(\frac{G(h)}{G(h_{ae})}\right)^\tau \left(\frac{1-|\alpha h|^{n-1}G(h)}{1-|\alpha h_{ae}|^{n-1}G(h_{ae})}\right)^\gamma, & h_j < h \le h_{ae} \\ 1, & h > h_{ae} \end{cases} \tag{6}$$


Obviously, the last term on the right-hand-side represents the hydraulic conductivity of the soil for liquid water. Here, $K_{s,c,jun}$, $\gamma$, and $\tau$ are fitting parameters that appear only in the expression for the UHCC, additional to $h_{ae}$, $\alpha$, and $n$, which also feature in the RIA model for the SWRC. If desired, some or all of the latter can be fitted independently from those of the SWRC. Because $K_{s,a}$ is not a fitting parameter, the junction model has one parameter less than the models that

average or add domain conductivities. In fact, it has the same number of parameters as the Kosugi model (de Rooij, 2024a) that does not consider adsorbed water. The value of $h_j$ follows from that of $h_d$ and $n$ according to Eq. (1c). Hence, only if $h_d$ and/or $n$ are fitted separately for the UHCC can $h_j$ differ from $h_{j\theta}$. The curve described by Eq. (6) has a discontinuous derivative at $h_j$, allowing it to reproduce the changing slope of the UHCC observed for some soils.





It is worth noticing that in the limit for $h \ll 0$, the following simplification holds for the wet branch of Eq. (6), 175 where $h_j < h \leq h_{ae}$.

$$\left(\frac{G(h)}{G(h_{ae})}\right)^{\tau} \left(\frac{1-|\alpha h|^{n-1}G(h)}{1-|\alpha h_{ae}|^{n-1}G(h_{ae})}\right)^{\gamma} \approx \left(\frac{h}{h_{ae}}\right)^{\tau(1-n)}, \ h \ll 0 \qquad (7)$$

This requires $\tau$ to be positive, which is stricter than the limits set by Peters et al. (2011) and Peters (2014). It is 180 physically plausible that the second factor on the left-hand side decreases as $h$ decreases. This is the case when $\gamma$ is positive.

For coding purposes, Eq. (6) and the corresponding expression for Kosugi's model as formulated by de Rooij (2024a) can be cast in a form that separates terms that need to be calculated once in case the SWRC parameters are fixed, terms that need to be calculated once for every iteration, and terms that need to be calculated for every iteration 185 and every matric potential corresponding to an observed point on the UHCC. The reformulated equations are given in Appendix A.

## 3. Fitting the model parameters

Before KRIAfitter 1.0 (de Rooij, 2024b, where the code, the User Manual, and example input and output files can be downloaded) can be run to determine the values of the parameters of the chosen UHCC model for a particular 190 soil, the parameters of the SWRC of Eqs. (1a–d) need to be fitted using RIAfitter 2.0 (de Rooij, 2024c, which also hosts its User Manual), or higher versions, once they become available. The water content at saturation has to be prescribed in the input for KRIAfitter because $\theta_s$ does not appear in the equations for any of the hydraulic conductivity models. The value of $h_d$ should not be fitted separately for the SWRC and the UHCC, because that would create a physically impossible situation where either a non-zero liquid-water conductivity exists although no liquid water is present, or 195 liquid water is present but is rendered immobile because the liquid-water hydraulic conductivity is zero. Hence, both $\theta_s$ and $h_d$ are fixed at the values of the SWRC that are provided on input.

Both KRIAfitter and RIAfitter are coded in Fortran2008, compiled on CygWin's Fortran compiler for 64-bit Windows computers, and tested on such a computer. All input and output consist of ASCII files. Typical run times vary from seconds to several minutes.

All hard-coded parameter values are placed in modules. This allows users to modify them as needed by changing a single value without having to inspect the entire code. The User Manuals list all hard-coded values and their respective module.





### 3.1 The parameter fitting algorithm

KRIAfitter fits up to six or seven parameters for one of six models for the unsaturated soil hydraulic
conductivity, depending on the model chosen by the user. For each of these models, the equivalent vapour conductivity
can be included or excluded, yielding twelve different conductivity models.

Each of $k$ observed data points provided on input needs to have an estimate of the standard deviation of the
measurement error of the matric potential or the volumetric water content, and of the associated soil hydraulic
conductivity. These are used to calculate the weighted squared error term $w_i$ assigned to the fit for data point $i$ as:


$$w_i = \left( \frac{K_{o,i} - K_{f,i}}{\frac{dK}{dx}\sigma_{x,i} + \sigma_{K,i}} \right)^2 \qquad (8)$$

where $K$ denotes the soil hydraulic conductivity (cm d$^{-1}$), $x$ denotes either the matric potential $h$ (cm H$_2$O) or the
volumetric water content $\theta$, $\sigma_x$ and $\sigma_K$ are the standard deviation of the measurement errors in the independent variable
($h$ or $\theta$) and in $K$, respectively. The subscript $o$ denotes an observed, and the subscript $f$ a fitted value. The subscript $i$
denotes the number of the observation, hence $i \in \{1,...,k\}$. The slope $dK/dx$ is evaluated at the observed value $x_{o,i}$ of the
independent variable and updated for every set of fitted parameter values.

The individual terms $w_i$ are used to compute the root mean square error (RMSE) of the fitted vs. the observed
values:


$$\text{RMSE} = \left[ \frac{\sum_{i=1}^{k} w_i}{\sum_{i=1}^{k} \left( \frac{dK}{dx}\sigma_{x,i} + \sigma_{K,i} \right)^{-1}} \right]^{\frac{1}{2}} \qquad (9)$$

In the code, the error standard deviations are scaled in such a way that the arithmetic mean of $\sigma_{K,i}$ equals 1% of the
largest conductivity value in the input file with the observation data. The required scale factor is applied to $\sigma_{x,i}$ as well
to conserve the relative weight of all observation errors.

The code minimizes the RMSE (the objective function) by the shuffled complex evolution algorithm (SCE). A
detailed description of the algorithm and discussions of its parameters are provided by Duan et al. (1992, 1993, 1994).
A brief summary of the SCE algorithm is provided in the User Manual (de Rooij, 2024b).

The number of dimensions of the parameter space (denoted NrOfDimensions in the code and below) is the
number of parameters whose values are actually fitted, as opposed to being set to a fixed value by the user. The number
of complexes (sets of points in the parameter space) adheres to Duan et al. (1994) for any number of fitting parameters,
but with a minimum of two, but only if input variable FewComplexes is set to 'T' on input. This results in two complexes





for any of the conductivity models. If FewComplexes is set to 'F', the number of complexes is $2 \times$ NrOfDimensions. Setting FewComplexes to 'F' significantly increases the execution time. Test calculations showed the quality of the fit

occasionally improves. Per the guidelines of Duan et al. (1994), the size of the complexes and the number of evolution steps between shuffles equal $(2 \times$ NrOfDimensions$) + 1$. The size of the subcomplexes is NrOfDimensions $+ 1$. The number of offspring in each evolution step is hard-coded as 1 (parameter SCEa in module SCEBisParameters).

Thus, the algorithm replaces the single poorest performing point of each subcomplex during an evolution step by a new point with a lower RMSE, or a random point if a better-performing point could not be found. It does so by

• first checking if a reflection point (further removed from the centroid of the subcomplex than the point that will be replaced) has a lower RMSE.

• If a parameter value of the reflection point is outside the permitted range, it is made equal to the minimum or maximum allowed value, whichever is closest. If the value of $h_j$ lies between $h_{ae}$ and zero, its value of $h_{ae}$ is increased to make $h_{ae}$ slightly larger (less negative) than $h_j$.

• If the reflection point does not perform better than the original point, a contraction point (closer to the centroid) is checked.

• If neither point improves the RMSE, a random point in the parameter space replaces the worst 'parent', even if its RMSE is larger.

After all evolution steps have been completed, all points in the complexes (including those that evolved in the

subcomplexes) are reshuffled and assigned to new complexes. The process of selecting subcomplexes and improving those through evolution steps is repeated for the new complexes. After each shuffle, the code checks for convergence. The code terminates when the parameter fits converged, or when the user-set maximum number of evaluations of the objective function has been exceeded. To check for convergence, the code evaluates the following convergence criteria for each fitting parameter after each shuffle. If none of the parameters fails more than the user-specified maximum

permitted number of criteria, convergence has been achieved.

1.  In the best fits from the last set of shuffles, the range of a parameter does not exceed both the absolute and the relative user-specified tolerance. The number of shuffles in the set is the maximum of two times the number of non-fixed fitting parameters and the minimum required number of shuffles (hard-coded parameter MinimumStoredShuffles).

2.  In the best fits from the last set of shuffles, the range of the objective function does not exceed its absolute user-specified tolerance.

3.  The parameter range in the final complexes does not exceed the maximum internally set permissible value.

4.  The volume of the hypercube enveloping the final complexes does not exceed the maximum internally set permissible value.



5. The parameter range in the most successful complex (minus the point with the highest RMSE) does not exceed the internally set maximum permissible value.

6. The volume of the hypercube enveloping the most successful complex (minus the point with the highest RMSE) does not exceed the internally set maximum permissible value.

7. A parameter does not exceed both the absolute and the relative user-specified tolerance in the final complexes.

8. A parameter does not exceed both the absolute and the relative user-specified tolerance in the most successful complex (without the point with the highest RMSE).

9. The change of the objective function between consecutive shuffles does not exceed the user-specified tolerance.

10. The RMSE of the fit does not exceed a user-specified tolerance.

Criteria 2, 4, 6, 9, and 10 apply to all parameters simultaneously. For large numbers of parameters, a hypercube that occupies only a small fraction of the hypercube defined by the ranges of the fitting parameters can still allow an excessively wide range for the individual parameters. The permissible volume therefore becomes smaller with an increasing number of fitting parameters. It is determined by a preset value (hard-coded parameter SCEHyperVolumeTolerance1D) raised to a power equal to the number of fitting parameters.

### 3.2 Validity of parameter values and their combinations

The SCE algorithm requires a finite parameter space in all dimensions, so permitted ranges of all parameters are required on input. The code checks the sign of all minimum and maximum values, corrects them if needed, and places the minimum and maximum value in correct order if needed. For $h_{ae}$, $\alpha$, $n$, $K_{s,c}$, $K_{s,a}$, $\gamma$, and $\tau$, physical or mathematical limits are implemented in subroutine RangeChecker, and the parameter range is forced to be within these bounds if needed. The routine also checks limits on $\theta_s$ and $h_d$, but since these parameters are fixed, these should be redundant. They are retained in case users wish to modify the code or use the routine for other purposes.

During the fitting process, for those models for which both $K_{s,a}$ and $K_{s,c}$ are fitting parameters, the code makes sure the former never exceeds the latter. Similarly, $h_j$ should not be closer to zero than $h_{ae}$. If it is, the value of $h_{ae}$ is modified. If that parameter is fixed, the value of $n$ is modified. Roundoff errors can create a combination of values for $h_d$, $h_j$, and $n$ that violate Eq. (1c). If that is the case, $n$ is modified as well.

### 3.3 Configuring the fitting process

If desired, the code generates a regular grid of map points with their associated RMSE that covers the parameter space. This can be arranged by setting Boolean input parameter UseMap to 'T'. The density of the grid is calculated from the maximum number of points supplied by the user, but a minimum number of points is generated that overrides a smaller user-supplied number. For parameters that are log-transformed, the map points are equidistant on the logarithmic scale. The map is used to modify the parameter space in order to focus on a hypercube



in which the lowest values of the RMSE are concentrated. If the code finds that these values occur close to a boundary of the parameter space, the range of a parameter can be expanded somewhat.

A small number of parameter optimization runs is performed (hard-coded as 3) within the updated parameter space. In optimization run 1, the first complexes are filled with points that have lowest RMSE values. The second run adds some random noise to the parameter values of these points to have slightly different initial complexes. The remaining run fills the complexes with randomly generated points.

If UseMap = 'F', a larger number of runs is performed (hard-coded as 50, see below), all of them with randomly generated points that fill the first complexes. This larger number of runs allows the calculation of statistics of the parameter value populations.

The user needs to specify for each parameter whether untransformed values or log-transforms of their absolute values (base 10) should be fitted. In the latter case, the corresponding dimension of the parameter space will be log-transformed as well, and the reflection and contraction steps are performed on log-transformed values. The fitted values that are written to output are back-transformed, but the statistics (mean, standard deviation, correlations and covariances) are for the log-transformed absolute values.

Any parameter can be fixed at a desired value by setting the upper and lower limit equal to that value. In that case, dummy values need to be provided for the absolute and relative tolerances, and for the Boolean variable that indicates if a log-transformation is needed. For the specific case that $h_{ae}$ and $h_j$ should be equal to those of the SWRC, $h_{ae}$ and $n$ will have to be fixed to the SWRC values. This can be achieved by setting input parameter AlphaOnly to 'T'. In that case, all input for $h_{ae}$ and $n$ is treated as dummy values, except for their identifying names that will be used in output.

Finally, the user can specify if the untransformed or log-transformed unsaturated soil hydraulic conductivity should be fitted by setting input parameter LogK to 'F' or 'T', respectively. In the latter case, both the observed and fitted conductivities are log-transformed before the RMSe is computed. If LogK = 'T', the error standard deviations for the observed conductivities apply to the log-transformed values. The User Manual offers some guidance on how to estimate these.

Generating a map of the RMSE in the parameter space (in case UseMap = 'T') of course increases computation time, especially if the number of dimensions is high. It can help constrain (or expand) the parameter space and give the initial complexes some guidance about the location of the global minimum. This can be advantageous if one is not sure about the range of the parameter values.

Executing many runs (in case UseMap = 'F') also increases processor time. If the output shows a large number of random points compared to the number of reflection and contraction points, it is possible that the minimum was found multiple times. If that is the case, it is recommended to increase the probability of convergence by allowing more



criteria to fail (the test runs gave good results for setting this number to 7). When doing so, the maximum number of iterations can also be reduced. If it is limited to 3000, most fits will probably still converge.

330        When a map is generated, the output includes the mean and standard deviation of the RMSE values of a random sample of the map points, as well as the cumulative probability density function of the RMSE in the parameter space and its histogram (also based on the sample). This information shows if there is a large portion of the parameter space in which the RMSE hardly varies, if there is a well-defined minimum, etc. For each run, correlation matrices of the parameter values are provided based on a sampling of the final iterations. These samples will include randomly

generated points, so the reported correlations will underestimate the true correlations.

        If no map is generated, the mean, population standard deviation, and histogram of all fitting parameters (or their log-transformed values) are written to output, as well as covariance and correlation matrices. These statistics are based on the fits of all runs, so they are expected to reflect the true values well. This information can be used, for instance, to sample the multivariate distribution of the parameter values in order to carry out a Monte Carlo study to

examine the effect of parameter uncertainty on the uncertainty of output of numerical models for unsaturated flow (de Jong van Lier et al., 2024).

        It may be useful to run KRIAfitter with UseMap = 'T', check the updated parameter ranges in the output (see the User Manual for details), copy these to the input file, and run KRIAfitter again with UseMap = 'F'. In this way, the parameter space is better defined, and one obtains reliable parameter statistics.

**4. Model evaluation**

**4.1 Fitting procedure**

        The first 13 soils in Table 1 of Weber et al. (2019) were used to evaluate the conductivity models. Instead of Ver P36 C1g in that table (P26 is a typo), we used UNSODA soil 4142, which refers to the same pair of samples. For all these soils, the data points span a wide range of matric potentials, and both retention and conductivity data are

available.

        First, the SWRC parameters were fitted, once with $h_d$ free, and once with $(1 + c)h_d$ fixed at $-10^{6.8}$ cm $H_2O$ (Schneider and Goss, 2012). Parameters $\alpha$ and $h_d$ were log-transformed because they can vary over several orders of magnitude (de Rooij, 2022). The error standard deviations were based on guestimates of the accuracy of observed water contents and matric potentials. These resulted in error bands $r$ with units depending on the variable. The

corresponding error standard deviation of $r/\sqrt{12}$ was found by assuming the error to be uniformly distributed within that band (Abramowitz and Stegun, 1964, p. 930). Because pressure plate data were found to be unreliable after the data sets were created (Bittelli and Flury, 2009), matric potentials below $-1000$ cm $H_2O$ were assigned an error band of $\pm0.1$ $h_o$ (with the subscript $o$ denoting an observed value).



The parameter space bounds were improved by fits that used the map of the RMSE, and then the improved
ranges were used to generate a fit without map. In a few cases (Pachappa, Rehovot sand, SM−1005, UNSODA 4650) the
data points were distributed so unevenly that the weights of the data points needed to be modified to compensate for
that. For all soils, the fit without map with $(1 + c) h_d$ fixed gave the lowest AICc value. Therefore, these parameter values
were used in the fits of the UHCC models. The fits also provided two estimates of the bulk soil hydraulic conductivity at
saturation, based on Eqs. (1) and (15) of Timlin et al. (1999). The latter estimate could only be calculated if $h_{ae} < 0$ and $n$
< 3 (de Rooij, 2024a). If the data range of conductivity data required it, these estimates were included in the file with
observation data and given the same error standard deviations as the other points.

Then, the five conductivity models of de Rooij (2024a) and the junction model (Eq. (6)) were fitted, with $\alpha$, $K_{s,c}$, and
$K_{s,a}$ log-transformed and the fits performed on log-transformed conductivity values. For each model, five sets of parameters
were fitted:

1.  all six or seven fitting parameters $h_{ae}$, $\alpha$, $n$, $K_{s,c}$, $K_{s,a}$ (when applicable), $\gamma$, and $\tau$ fitted
2.  $h_{ae}$, $h_j$, and $h_d$ as for the SWRC; $\alpha$, $K_{s,c}$, $K_{s,a}$ (when applicable), $\gamma$, and $\tau$ fitted
3.  $\tau = 0.0$ (Assouline, 2001); $h_{ae}$, $\alpha$, $n$, $K_{s,c}$, $K_{s,a}$ (when applicable), and $\gamma$ fitted
4.  $\gamma = 2.0$, $\tau = 0.5$ (Mualem, 1976); $h_{ae}$, $\alpha$, $n$, $K_{s,c}$, and $K_{s,a}$ (when applicable) fitted
5.  $h_{ae}$, $\alpha$, and $n$ as for the SWRC; $K_{s,c}$, $K_{s,a}$ (when applicable), $\gamma$, and $\tau$ fitted

As was the case for the SWRC, these fits were first performed using the map of the RMSE to better define parameter ranges.
Then the fit was repeated with the improved ranges. If the upper limit of $K_{s,a}$ exceeded the lower limit of $K_{s,c}$, it was capped at
the value of the latter. Up to seven of the ten convergence criteria were allowed to be missed, and the maximum number of
evaluations of the objective function was 3000. The error standard deviations were estimated for the largest observed
conductivity only (see the User Manual for details), and then assigned to all data points so that the data were all weighted
equally. The only exception was soil UNSODA 4010, where the first three observations and the two estimates of the saturated
bulk conductivity $K_s^B$ had to be given more weight to obtain good fits. Vapour flow was always included. All data were from
laboratory measurements, so the temperature was set at 20°C.

This required 780 runs of KRIAfitter, in which 20670 fits were performed in total (3 fits per run with map, 50
without). In the vast majority of cases, fits converged after several hundreds of evaluations of the objective function.
The best fits of the runs without map will be discussed below.

## 4.2 Fitting results

The RIA parameterization (Eqs. (1a–d)) fits the intermediate and wet range quite well for a wide range of
shapes of the curves, as the graphs in the Appendix B show. Because of the lower weight and the pF of oven-dryness
fixed at 6.8, the fitted dry branches are drier than the observations to varying degrees, consistent with the expectation
that pressure plate data tend to underestimate $|h|$. For Pachappa, the fitted value of $h_{ae}$ is −15.6 cm $H_2O$, which seems

a bit arbitrary, but with data points missing for matric potentials between 0.0 and −47.0 cm H₂O, the code has insufficient information for a reliable estimate. The desirability of additional points to guide the estimation of $h_{ae}$ also seems apparent to a more moderate degree for SM−1005 and UNSODA 2571.

Appendix B also shows the 30 fits of the UHCC for each soil, separated into one graph each for every of the five sets of fitting parameters listed above. The curves are arranged in the same order as that list when read from left-to-right, top-to-bottom. The main findings from a visual analysis of the collection of graphs are presented below. For consistency between this text, Appendix B, and the User Manual, the various conductivity models will be referred to by their identifying three-character labels as follows:

- ADV: the unweighted additive model, with vapour flow (de Rooij, 2024a)
- AMV: the arithmetic mean model, with vapour flow (de Rooij, 2024a)
- GMV: the geometric mean model, with vapour flow (de Rooij, 2024a)
- HMV: the harmonic mean model, with vapour flow (de Rooij, 2024a)
- JUV: the junction model, with vapour flow (Eq. (6))
- KGV: the Kosugi model with capillary water only, with vapour flow (de Rooij, 2024a)

Pachappa is the only soil for which $K(\theta)$ was fitted instead of $K(h)$. Despite the number of fitting parameters ranging from 3 to 7, the fits show only minor variations, limited to the dry range, which has a single data point.

Pachepsky has $K(h)$ data over a wide range, but not at saturation. In view of the small amount of noise in the data, $K_{s,c}$ and $K_{s,a}$ are both fitted without providing estimates of saturated bulk soil hydraulic conductivity $K_s^B$ (LT⁻¹). The estimates based on the fitted SWRC are 182 and 110 cm d⁻¹. For the fit with all SWRC parameters fixed (set nr. 5 in the list above), all fitted saturated conductivities are well below these, and often even below observed values below saturation. For fitting parameter sets 1 through 4, only JUV and KGV have plausible values of $K_s^B$. GMV and HMV underperform overall. JUV gives the best overall fits when all parameters are fitted (set 1) or $\tau$ is fixed at zero (set 3).

The UHCC for Rehovot sand has a steep descent that transitions abruptly into a gentler slope. This shape can only be reproduced by ADV and AMV, and when all 7 parameters are fitted (set 1) or $\tau$ is fixed at zero (set 3).

The shape of SM−6−62 is accurately fitted by ADV, AMV, and JUV if 6 to 8 parameters can be fitted. All do well for set 1 (all parameters free) and set 3 (one parameter fixed). ADV and AMV also give good fits for set 4 (with $\gamma$ and $\tau$ fixed to Mualem's (1976) values, leaving only four fitting parameters for JUV), although the conductivity may stay high for too long as the soil desaturates.

The fits for SM−22−88 are best for sets 1 and 3. JUV does particularly well if all parameters are free (set 1), AMV and ADV have an extra parameter and still do well if $\tau$ is fixed (set 3).

The UHCC of SM−35−119 is fitted well by all conductivity models for all sets except set 4 (all SWRC parameters fixed). For set 1, the spread between the fitted curves is remarkably high in the data-free wet range.





For SM−41−127, ADV, AMV, JUV, and KGV all do well if all parameters are free (set 1). For the other sets, the somewhat isolated driest point is only fitted well by ADV and AMV. The observation range covers only a narrow pF–
range, resulting in considerable spread outside this range. The spread is contained at saturation by the values estimated from the corresponding SWRC.

The data points for SM−1005 are grouped in two clusters. HMV and GMV consistently underperform. ADV and JUV give good fits for up to one fixed parameter (sets 1 and 3), and because of its extra parameter, ADV still does well with two fixed parameters (set 4). Interestingly, AMV gives a curvy fit for set 4 that matches the data well.

The UHCC of UNSODA 2571 is linear on the double-log scale and fitted well for all sets by all models except HMV. The spread in the dry range is large, with ADV and AMV separated from the other models.

The peculiar shape of the UHCC of UNSODA 4010 cannot be fitted properly by any model. ADV, JUV, and KGV give somewhat acceptable fits for set 2 (with $\alpha$ the only free SWRC parameter).

The wet range of the UHCC of UNSODA 4031 is fitted poorly by HMV and especially GMV. For sets 1 and 3, all
other models give good fits over the entire data range, with limited spread outside that range.

For UNSODA 4142, most models give good and comparable fits for sets 1–3, except AMV, GMV, and HMV in the wet range for set 1, and AMV and GMV for set 3. ADV tends to give higher conductivities in the dry range than the other models. The fits of JUV and KGV are nearly identical for all sets.

None of the fits for UNSODA 4650 are very good because the data points appear in two disjointed clusters with
a discontinuity between them that can only be somewhat captured by ADV and AMV.

In many cases, GMV and HMV underestimate the conductivity in the wet range. This is related to the effect of the value of $K_a$ (much lower than $K_c$) on the geometric and harmonic means of the two. Even though the upper limit of $K_{s,c}$ was permitted to be about 20 times the measured saturated conductivity during the fitting process, this often was still insufficient to let the mean reach the observed values.

In 75% of the plots, KGV closely tracks JUV. For a model that lacks representation of water adsorbed in films, KGV performs remarkably well. AMV overlaps ADV in the wet range at times, but the curves tend to separate in the dry end. ADV often has the highest conductivity in the dry range of all models, until they all coalesce on the vapour conductivity, which is the only non-zero component of the bulk conductivity for $h \leq -10^{6.8}$ cm $H_2O$, where liquid water ceases to exist.

JUV and KGV have one parameter less than the other models, yet JUV in particular tends to be among the best overall fits for a given soil based on the visual inspection of the full set of graphs. If it does so even when sharing the top spot with other models in the same set (e.g. Pachepsky, SM−35−119, SM−1005), this means it can match the performance of other models with fewer parameters. This suggests that ADV and the averaging models may be overparameterized, especially if all their seven parameters are fitted. On the other hand, for the Rehovot sand, ADV and
AMV are the only models that are versatile enough to produce good fits, so the extra parameter may be needed at times.



Although AMV and ADV often diverge in the dry range, AMV does not appear to substantially increase the fitting prowess provided by the combination of JUV and ADV. From the fits it appears that with only ADV and JUV, the best possible result (based on visual inspection) can be found for all soils used in the test.

This can be explored further through the goodness of fit measures and parameter correlation matrices. The
time needed do so for all 390 fits presented in Appendix B is prohibitive. Instead, of the thirty combinations of conductivity model and fitting parameter set for each soil, the ones that give the smallest values of the RMSE and of Akaike's Information Criterion corrected for small sample sizes (Hurvich and Tsai, 1989; denoted AICc) are identified. Only soils for which the visual analysis showed that both the SWRC and the UHCC were fitted well at least once are included in Table 1. The soils not considered are Pachappa (questionable value of $h_{ae}$ in the SWRC due to lack of data
in the wet range), UNSODA 4010, and UNSODA 4650 (both because none of the combinations could fit the conductivity curve well).

| Table 1. Combinations of conductivity models and fitted parameter sets (see main text for the parameter sets corresponding to the listed numbers) that give the lowest Root Mean Square Errors (RMSE) and values of the corrected Akaike's Information Criterion (AICc) for soils with acceptable fits of both the soil water retention curve and the unsaturated hydraulic conductivity curve. | | |
|---|---|---|
| Soil | Model–parameter set combination that gives the lowest value of the indicated criterion | |
| | RMSE | AICc |
| Pachepsky | ADV, 1 | ADV, 2 |
| Rehovot sand | AMV, 1 | AMV, 3 |
| SM−6−62 | ADV, 1 | ADV, 1 |
| SM−22−88 | ADV, 1 | ADV, 4 |
| SM−35−119 | JUV, 3 | JUV, 3 |
| SM−41−127 | AMV, 1 | AMV, 4 |
| SM−1005 | JUV, 1 | JUV, 1 |
| UNSODA 2571 | AMV, 1 | JUV, 5 |
| UNSODA 4031 | JUV, 1 | JUV, 4 |
| UNSODA 4142 | ADV, 1 | ADV, 4 |

Based on the minimal RMSE, ADV performs best for four soils, AMV and JUV for three. The most flexible
parameter set (nr. 1) prevails nine times, the second-most flexible one (nr. 3), one time. But if the required number of fitting parameters is accounted for by using the AICc instead of the RMSE, ADV and JUV both achieve the minimum AICc





four times, and AMV twice. Set 4, with $\gamma$ and $\tau$ fixed to Mualem's (1976) values, performs best four times, sets 1 and 3 twice, and sets 2 and 5 once.

A more detailed analysis is carried out for a subset of soils. Based on Table 1, only ADV and JUV are considered,
as they are the overall best-performing models, with KGV included to see to what degree the more advanced models outperform the classical model. This analysis serves two purposes: an evaluation of conductivity models and fitting parameters sets, and a demonstration of ways to interrogate the output of KRIAfitter.

The members of this subset of soils must have a good fit of the SWRC, so that spillover effects from that fit do not cloud the assessment of the conductivity models. The subset must also represent different shapes of the SWRC and
the UHCC, and different qualities of conductivity data. Pachepsky's soil has a sigmoidal SWRC and conductivity data that cover a wide range. SM−35−119 has an SWRC with three linear branches and conductivity data over a narrow range, with estimated saturated conductivity values. UNSODA 2571 has a power-law type SWRC (Brooks and Corey, 1964) and conductivity data over an intermediate range. The subset is completed by Pachappa, because it is the only soil for which the conductivity data have $\theta$ as the independent variable.

The values of the RMSE and AICc of each of the fits discussed above are presented in Tables 2–5. If only one of two identical values is designated optimal in these tables, they are different when more significant digits are considered.

Values of AICc should never be compared between soils. Because the error standard deviations are scaled by the code using the maximum conductivity value in the input file with the measurements (see the User Manual), RMSE
values for different soil cannot be compared either.

Table 2. Root Mean Square Errors (RMSE) and values of the corrected Akaike's Information Criterion (AICc) for additive (ADV), junction (JUV), and capillary water only (KGV) conductivity models, fitted to data for Pachappa. Best values for a particular conductivity model are in italics. Overall best values are bold.

| Fitted set nr. | RMSE | | | AICc | | |
|---|---|---|---|---|---|---|
| | ADV | JUV | KGV | ADV | JUV | KGV |
| 1 | 0.05141 | 0.05141 | 0.05141 | 33.71 | −2.959 | −2.959 |
| 2 | 0.16318 | 0.16248 | 0.16243 | 4.116 | −6.985 | −6.985 |
| 3 | **0.05121** | *0.05122* | *0.05121* | −3.045 | −21.38 | −21.38 |
| 4 | 0.05751 | 0.05782 | 0.05782 | *−18.83* | **−29.71** | **−29.71** |
| 5 | 0.16148 | 0.16382 | 0.16382 | −7.114 | −14.13 | −14.13 |





| Table 3. Similar to Table 1, but for Pachepsky's soil. | | | | | | |
|---|---|---|---|---|---|---|
| Fitted set nr. | RMSE | | | AICc | | |
| | ADV | JUV | KGV | ADV | JUV | KGV |
| 1 | **0.16141** | 0.17154 | 0.35483 | −44.37 | −45.45 | −7.656 |
| 2 | 0.16157 | 0.39162 | 0.39055 | **−52.37** | −9.748 | −9.891 |
| 3 | 0.16164 | *0.17152* | *0.35177* | −48.54 | *−49.26* | −11.91 |
| 4 | 0.17625 | 0.27293 | 0.36717 | −47.84 | −28.52 | *−13.10* |
| 5 | 0.19529 | 0.46275 | 0.46215 | −45.93 | −41.65 | −4.232 |

| Table 4. Similar to Table 1, but for SM−35−119. | | | | | | |
|---|---|---|---|---|---|---|
| Fitted set nr. | RMSE | | | AICc | | |
| | ADV | JUV | KGV | ADV | JUV | KGV |
| 1 | 0.00933 | 0.00791 | *0.00795* | −137.9 | −150.3 | −150.1 |
| 2 | 0.01934 | 0.05568 | 0.05425 | −119.4 | −81.24 | −82.28 |
| 3 | *0.00887* | **0.00789** | 0.00796 | −145.7 | **−155.2** | *−154.9* |
| 4 | 0.00925 | 0.03033 | 0.01561 | *−148.9* | −105.5 | −132.1 |
| 5 | 0.03274 | 0.11391 | 0.11256 | −102.5 | −56.23 | −56.70 |

| Table 5. Similar to Table 1, but for UNSODA 2571. | | | | | | |
|---|---|---|---|---|---|---|
| Fitted set nr. | RMSE | | | AICc | | |
| | ADV | JUV | KGV | ADV | JUV | KGV |
| 1 | 0.06683 | *0.08567* | *0.08666* | Invalid | ∞ | ∞ |
| 2 | 0.08050 | 0.08976 | 0.08976 | 63.69 | 9.429 | 9.429 |
| 3 | **0.04128** | 0.08881 | 0.08880 | ∞ | 65.26 | 65.26 |
| 4 | 0.07918 | 0.08883 | 0.08882 | 63.42 | 9.263 | 9.262 |
| 5 | 0.08188 | 0.09421 | 0.09421 | *7.961* | **−8.462** | *−8.462* |


Because ADV has one fitting parameter more than both JUV and KGV, it achieves the lowest RMSe for three out of four soils, albeit with a shared first place for Pachappa. JUV has the lowest RMSE for SM−35−119. The three RMSE values for Pachappa are nearly identical and all are achieved by set 3, with $\tau$ fixed at 0.0 (Table 2). For each soil, the lowest RMSE is achieved by either fitting set 1 (one case: Pachepsky; Table 3) or set 3 (three cases, Tables 2, 3, 5).



500         The more parsimonious sets of fitting parameters not surprisingly tend to achieve the smallest AICc values: sets 2, 3, 4, and 5 all achieve the lowest AICc for one soil. Note that UNSODA 2571 (Table 5) has only 8 data points, which means that for 6 fitting parameters, AICc becomes infinite, and beyond that, incalculable (see the User Manual and Hurvich and Tsai (1989) for details). JUV is the most successful conductivity model in terms of minimizing AICc, with KGV close behind. JUV has the lowest AICc for Pachappa with set 4, for SM−35−119 with set 3, and for UNSODA

2571 with set 5 (Tables 2, 4, 5). In all cases, KGV performs equally well or slightly worse. On average, 6 parameters are needed to achieve the lowest RMSE, and 4.25 to achieve the lowest AICc.

        Figure 1 shows the pairs of best fits based on RMSE and AICc values. Only the case for which the difference in the number of fitting parameters between RMSE and AICc-based fits is larger than two results in significantly different fits. The other three are indistinguishable or, in one case, identical.



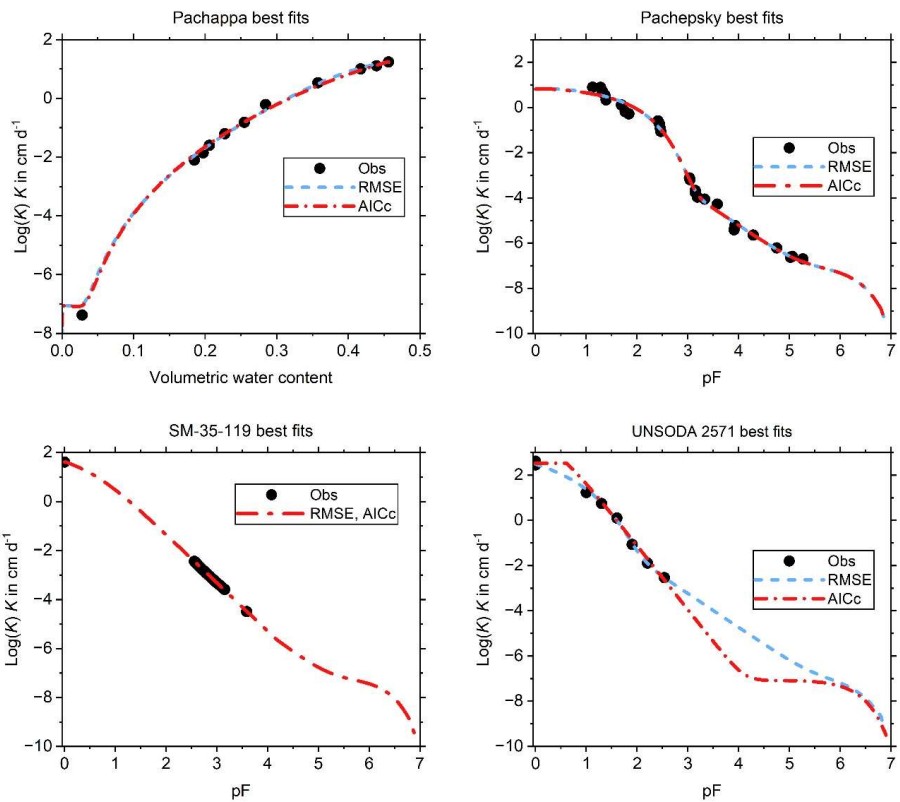


**Figure 1:** The fits that had either the smallest value of the Root Mean Square Error (RMSE) or Akiake's Information Criterion corrected for small samples (AICc) of all fits, i.e. those fits that are identified by the bold values in Tables 2–5.

The output of KRIAfitter when no map of the RMSE is generated permits an inspection of the correlation

structure. The breadth of parameter correlations can be assessed by determining from the correlation matrix in file SCEfittingK.OUT the number of data pairs for which the absolute value of the correlation coefficients exceeds a threshold. Table 6 shows the results for this threshold set to 0.8. There is no consistency for any of the conductivity models or fitted parameter sets between soils. Apparently, the properties of the input data set have far more influence.



Table 6. Number of data pairs with $|R| > 0.8$ for selected soils, conductivity models, and fitted parameter sets (see main text).

| Soil | Fitted set nr. | Conductivity model | | |
|---|---|---|---|---|
| | | ADV | JUV | KGV |
| Pachappa | 1 | 1 | 2 | 2 |
| | 2 | 1 | 1 | 1 |
| | 3 | 3 | 2 | 1 |
| | 4 | 6 | 1 | 1 |
| | 5 | 0 | 0 | 0 |
| Pachepsky | 1 | 0 | 1 | 1 |
| | 2 | 3 | 2 | 1 |
| | 3 | 2 | 1 | 1 |
| | 4 | 2 | 0 | 1 |
| | 5 | 1 | 0 | 0 |
| SM−35−119 | 1 | 0 | 0 | 0 |
| | 2 | 0 | 1 | 1 |
| | 3 | 0 | 1 | 1 |
| | 4 | 0 | 1 | 1 |
| | 5 | 0 | 1 | 0 |
| UNSODA 2571 | 1 | 0 | 0 | 0 |
| | 2 | 1 | 1 | 1 |
| | 3 | 0 | 1 | 1 |
| | 4 | 1 | 1 | 2 |
| | 5 | 1 | 1 | 1 |


Table 7 lists the correlation coefficients ($R$) with the maximum absolute value. Intriguingly, the only consistency here appears to be that the sets with the largest (set 1) and smallest (set 5) number of fitting parameters have the fewest data pairs with very high correlation coefficients ($R^2 > 0.9$, corresponding to $|R| > 0.95$). For set 5, this is understandable: with three (JUV, KGV) or four (ADV) parameters resulting in three or six possible parameter pairs,
there is limited opportunity for different parameter combinations that give the same RMSE across a range of values. And these will only give high $R$-values if the resulting relationship between the parameter values is linear.



Table 7. The maximum value of $|R|$ for any parameter pair for selected soils, conductivity models, and fitted parameter sets (see main text). Values that give $R^2 > 0.9$ are in italics.

| Soil | Fitted set nr. | Conductivity model | | |
|---|---|---|---|---|
| | | ADV | JUV | KGV |
| Pachappa | 1 | −0.8314 | 0.8457 | 0.8593 |
| | 2 | −0.9384 | *−0.9641* | *−0.9641* |
| | 3 | *−0.9698* | *−0.9909* | *−0.9543* |
| | 4 | *−0.9999* | *−0.9575* | *−0.9575* |
| | 5 | 0.7061 | 0.6658 | 0.6658 |
| Pachepsky | 1 | 0.7377 | −0.8471 | −0.9264 |
| | 2 | *−0.9922* | *−1.0000* | *−0.9999* |
| | 3 | *−0.9937* | −0.8818 | *−0.9573* |
| | 4 | −0.8525 | 0.6331 | 0.8486 |
| | 5 | 0.8638 | −0.5860 | 0.1933 |
| SM−35−119 | 1 | 0.6823 | −0.6732 | −0.6346 |
| | 2 | 0.4730 | *−1.0000* | *−1.0000* |
| | 3 | 0.5543 | *−0.9520* | *−0.9693* |
| | 4 | −0.4134 | −0.8938 | −0.8941 |
| | 5 | 0.7729 | *0.9703* | 0.3751 |
| UNSODA 2571 | 1 | −0.7285 | −0.6723 | −0.7554 |
| | 2 | −0.8201 | *−1.0000* | *−1.0000* |
| | 3 | −0.7916 | *−0.9731* | *−0.9659* |
| | 4 | −0.9047 | −0.8784 | −0.8758 |
| | 5 | *−0.9951* | −0.8032 | −0.8426 |

Finally, Table 8 gives the parameter pairs for which $R^2 > 0.9$. Here too, the dominant effect of the input data is
apparent, but there also appears to be an effect of the set of fitting parameters. Parameter pair $(\alpha, \tau)$ appears six times, five of which are for Pachappa. The three appearances of $(\alpha, n)$ are also for Pachappa, and for fitting parameter set 4. Seven of the eight appearances of $(\gamma, \tau)$ are for set 2, and all four appearances of $(n, \gamma)$ are for set 3, for models JUV or KGV. All things considered, no general guidelines can be given. Parameter correlations therefore need to be examined



on a case-by-case basis. If highly correlated parameters are found, the means and standard deviations of the individual
parameters that are also provided in file SCEfittingK.OUT can help determine a suitable value at which to fix one of the
parameters, if so desired.

Table 8. The parameter pairs that give $R^2 > 0.9$ pair for selected soils, conductivity models, and fitted parameter sets (see main text).

| Soil | Fitted set nr. | Conductivity model | | |
|---|---|---|---|---|
| | | ADV | JUV | KGV |
| Pachappa | 1 | — | — | — |
| | 2 | — | $\alpha, \tau$ | $\alpha, \tau$ |
| | 3 | $\alpha, \tau$ | $\alpha, \tau$ | $\alpha, \tau$ |
| | 4 | $\alpha, n$ | $\alpha, n$ | $\alpha, n$ |
| | 5 | — | — | — |
| Pachepsky | 1 | — | — | — |
| | 2 | $\gamma, \tau$ | $\gamma, \tau$ | $\gamma, \tau$ |
| | 3 | $\alpha, \tau$ | — | $n, \tau$ |
| | 4 | — | — | — |
| | 5 | — | — | — |
| SM−35−119 | 1 | — | — | — |
| | 2 | — | $\gamma, \tau$ | $\gamma, \tau$ |
| | 3 | — | $n, \gamma$ | $n, \gamma$ |
| | 4 | — | — | — |
| | 5 | — | $K_{s,a}, \tau$ | — |
| UNSODA 2571 | 1 | — | — | — |
| | 2 | — | $\gamma, \tau$ | $\gamma, \tau$ |
| | 3 | — | $n, \gamma$ | $n, \gamma$ |
| | 4 | — | — | — |
| | 5 | $\gamma, \tau$ | — | — |



### 5. Conclusions

The graphs in Appendix B, the combinations of models and fitted parameter sets that minimize AICc in Table 1, and the quality of the fits in Fig. 1 all indicate that all UHCC curves that can be fitted by at least one of the full set of six models, can probably also be fitted by ADV and/or JUV, without the need for implausibly high values of $K_{s,c}$ that arise from averaging hydraulic conductivities.

Many of the fitted curves depicted in Appendix B are remarkably similar, despite different numbers of fitted parameters. Only three out of ten soils in Table 1 have the same combination minimizing both the RMSe and AICc. In only three cases in Table 1 is the AICc minimal for a model with six or more parameters. The two upper top panels of Fig. 1 have indistinguishable plots based on fits with different numbers of parameters. All this suggests that conductivity models with six or seven fitting parameters are overparameterized for the UHCC of many soils. The

junction model and the classical Kosugi model have one parameter less than the other models tested in this study, and cannot have more than six. Of the two, only the junction model has the ability to reduce its slope abruptly in the dry range (best visible for the UHCC plots for soils SM−6−62, SM−22−88, and SM−1005 in Appendix B). Of the models that incorporate different domains of water in the soil, the junction model is the only one that sidesteps the fundamental impossibility of correctly averaging or adding domain conductivities (de Rooij, 2024a). Together with its parsimony,

this makes the junction model an attractive alternative to existing conductivity models.

### Appendix A

The expressions for the unsaturated hydraulic conductivity have terms and factors that only depend on the parameters of the SWRC. Other factors depend on the conductivity parameters. Finally, there are terms and factors that depend on the matric potential.

The efficiency of the parameter fitting algorithm for the case when all SWRC parameters are fixed was improved by taking this into account. The equations of the conductivity models were rearranged into separate terms and factors. Below, the rearranged equations are given. In the equations, the continuity correction factor $c$ is implemented. The symbols for the new factors (two characters and a number) reflect their names in the parameter estimation code, except for KG5, which is termed SWRCGhae in the code.

For Kosugi's (1999) model (de Rooij, 2024a) and the additive model (de Rooij, 2024a), the rearranged equations are as follows.





$$K(h) = K_s \cdot \begin{cases} 0, & h \leq (1+c)h_d \\ \left\{\beta\ln\left[\frac{(1+c)h_d}{h}\right]\right\}^\tau \left(\frac{KG1 - \frac{\beta}{|h|}}{KG4}\right)^\gamma, & (1+c)h_d < h \leq h_j \\ \left(\frac{G(h)}{KG5}\right)^\tau \left(\frac{KG2 + \frac{F(h) - KG3}{KG5}}{KG4}\right)^\gamma, & h_j < h \leq h_{ae} \\ 1, & h > h_{ae} \end{cases}$$

(A1a)

$F(h) = \frac{|\alpha h|^n G(h)}{|h|}$

(A1b)

$KG1 = \frac{\beta}{(1+c)|h_d|}$

(A1c)

$KG2 = KG1 - \frac{\beta}{|h_j|}$

(A1d)


$KG3 = \frac{|\alpha h_j|^n G(h_j)}{|h_j|}$

(A1e)

$KG4 = KG2 + \frac{|\alpha h_{ae}|^n}{|h_{ae}|} - \frac{KG}{G(h_{ae})}$

(A1f)

$KG5 = G(h_{ae})$

(A1g)

During parameter fitting, $KG1$ through $KG5$ only need to be calculated once if the SWRC parameters are fixed.

For the junction model of Eq. (6) of the main text, the rearranged equations are as follows. The hydraulic conductivity for liquid water is denoted $K_{l,jun}$.


$$K_{l,jun}(h) = \begin{cases} 0, & h \leq (1+c)h_d \\ JU4\left(\frac{h}{h_j}\right)^{-1.5}, & (1+c)h_d < h \leq h_j \\ K_{s,c}\left(\frac{G(h)}{G(h_{ae})}\right)^\tau \left(\frac{1 - |\alpha h|^{n-1} G(h)}{JU}\right)^\gamma, & h_j < h \leq h_{ae} \\ K_{s,c}, & h > h_{ae} \end{cases}$$

(A2a)

$JU1 = K_{s,c}\left(\frac{G(h_j)}{G(h_{ae})}\right)^\tau$

(A2b)



$JU2 = 1 - |\alpha h_j|^{n-1} G(h_j)$ (A2c)

$JU3 = 1 - |\alpha h_{ae}|^{n-1} G(h_{ae})$ (A2d)

$JU4 = JU1 \left(\dfrac{JU2}{JU3}\right)^{\gamma}$ (A2e)


$JU2$ and $JU3$ need only be calculated once if the SWRC parameters are fixed, $JU1$ and $JU4$ for every iteration of the fitting process. If at least one of the SWRC parameters are fitted, all four terms have to be recalculated one per iteration.

**Appendix B**

This appendix shows the graphs of all fitted SWRCs and UHCCs. The SWRC has a single fit for each of the test
soils (Fig. B1). The UHCCs of six conductivity models were fitted for five sets of fitting parameters, resulting in 30 fits for each soil (Figs. B2–B14). In the figure panels, the fitting parameter sets are labeled 'all free', 'alpha only', 'Assouline', 'Mualem', and 'SWRC fixed', corresponding to sets 1…5 in the main text. For Pachappa, the unsaturated hydraulic conductivity was observed as a function of the volumetric water content, for all other soils as a function of the matric potential.



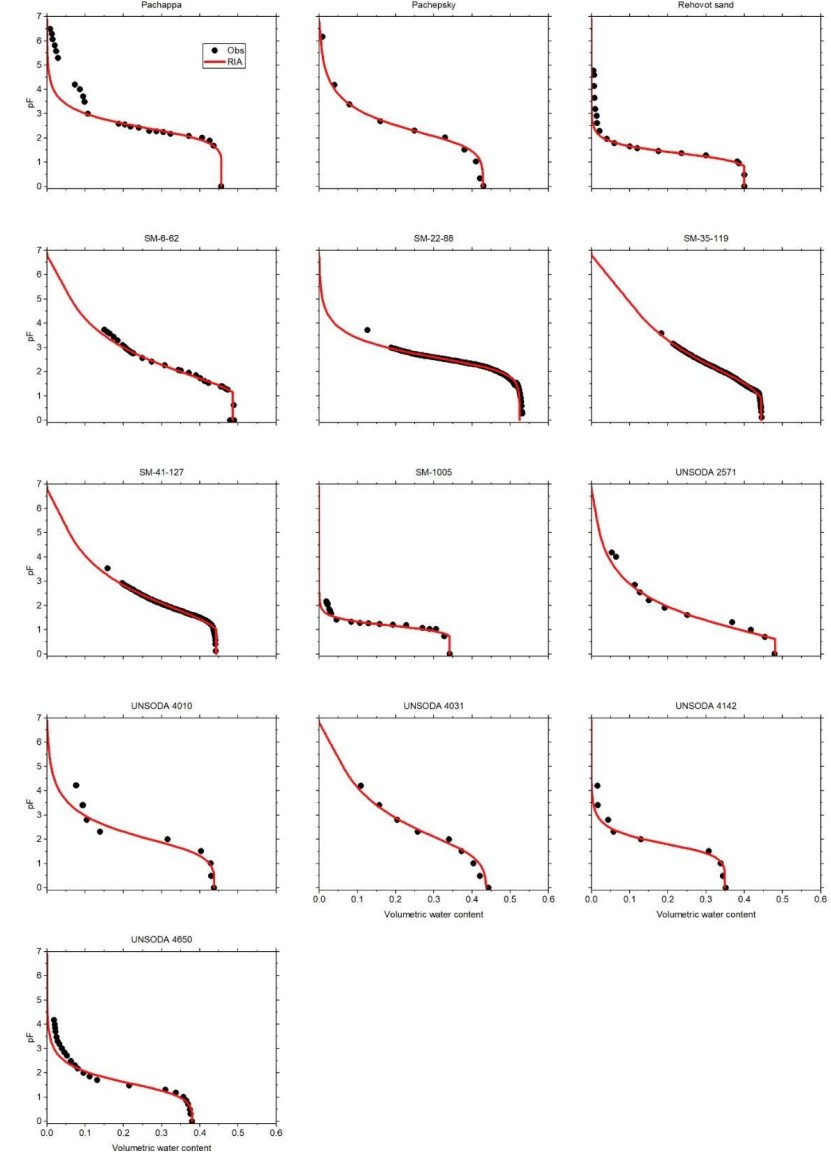


Figure B1: The observed and fitted soil water retention curves for 13 soils listed by Weber et al. (2019). The fits all had $(1+c)h_d$ fixed at $-10^{6.8}$ cm $H_2O$. The data points for pF > 3 were given less weight to reflect the low reliability of pressure plate retention data (Bittelli and Flury, 2009).



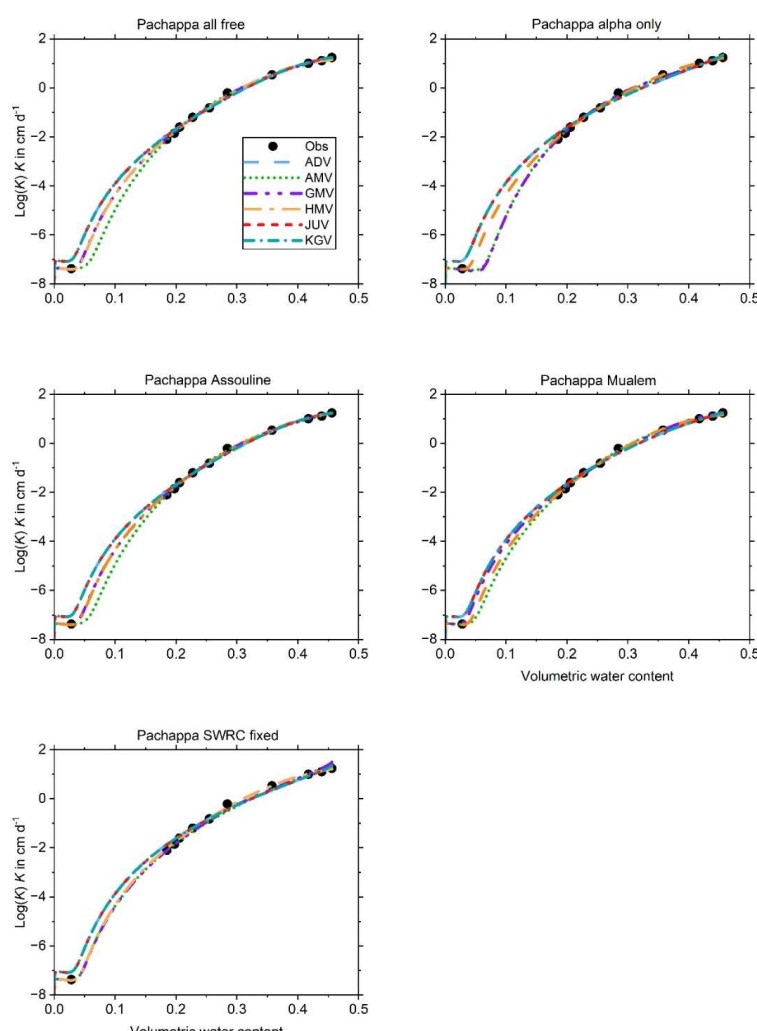


**Figure B2:** Fitted unsaturated hydraulic conductivity curves (UHCCs) for six models and five set of fitting parameters (both explained in the main text) for the Pachappa soil of Weber et al. (2019).



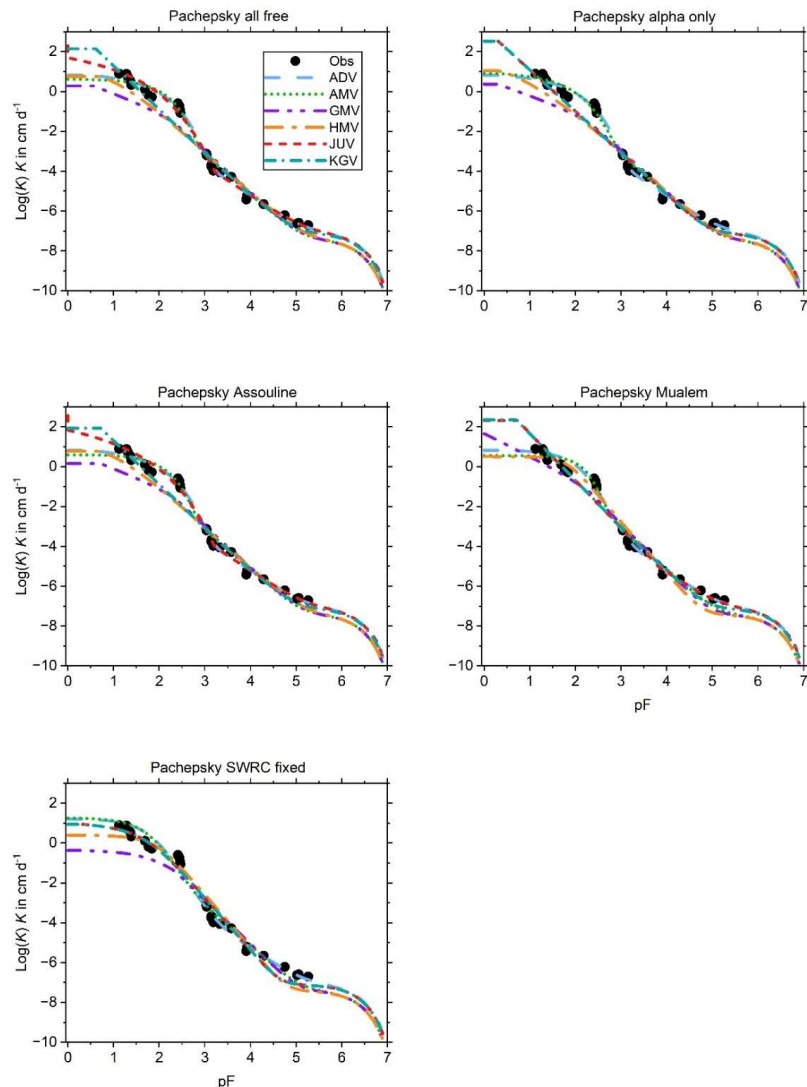

**Figure B3: As Fig. B2, but for Pachepsky's soil.**



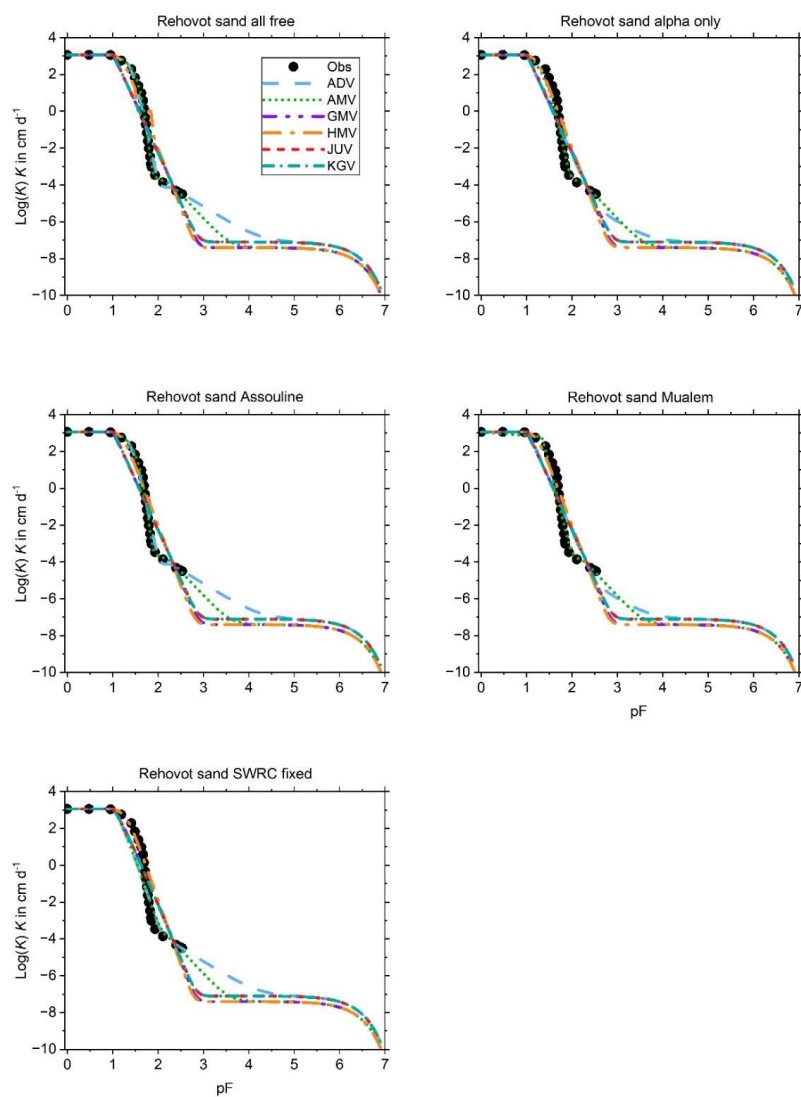


**Figure B4: As Fig. B2, but for Rehovot sand.**



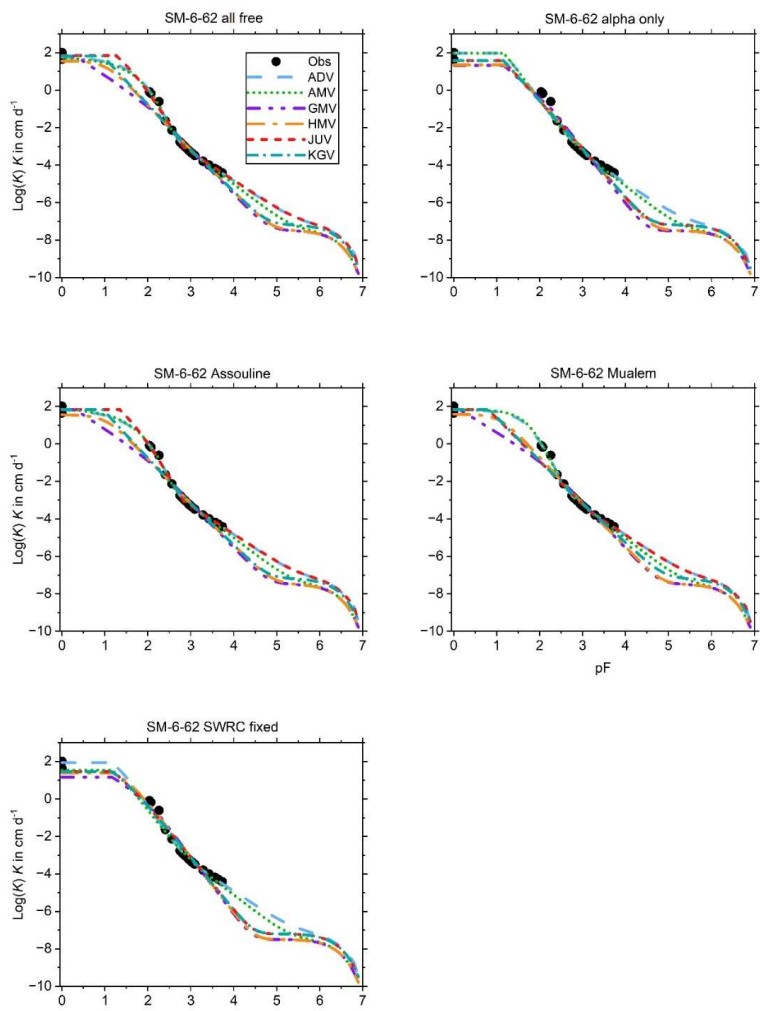

Figure B5: As Fig. B2, but for SM–6–62.



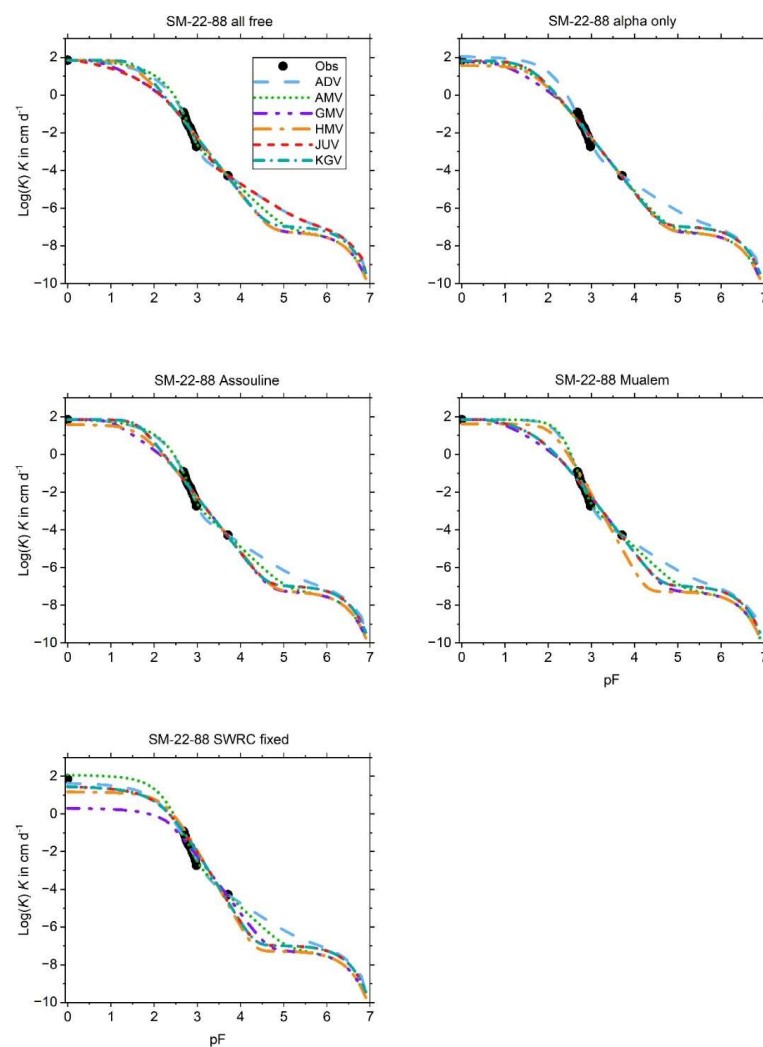


**Figure B6: As Fig. B2, but for SM–22–88.**





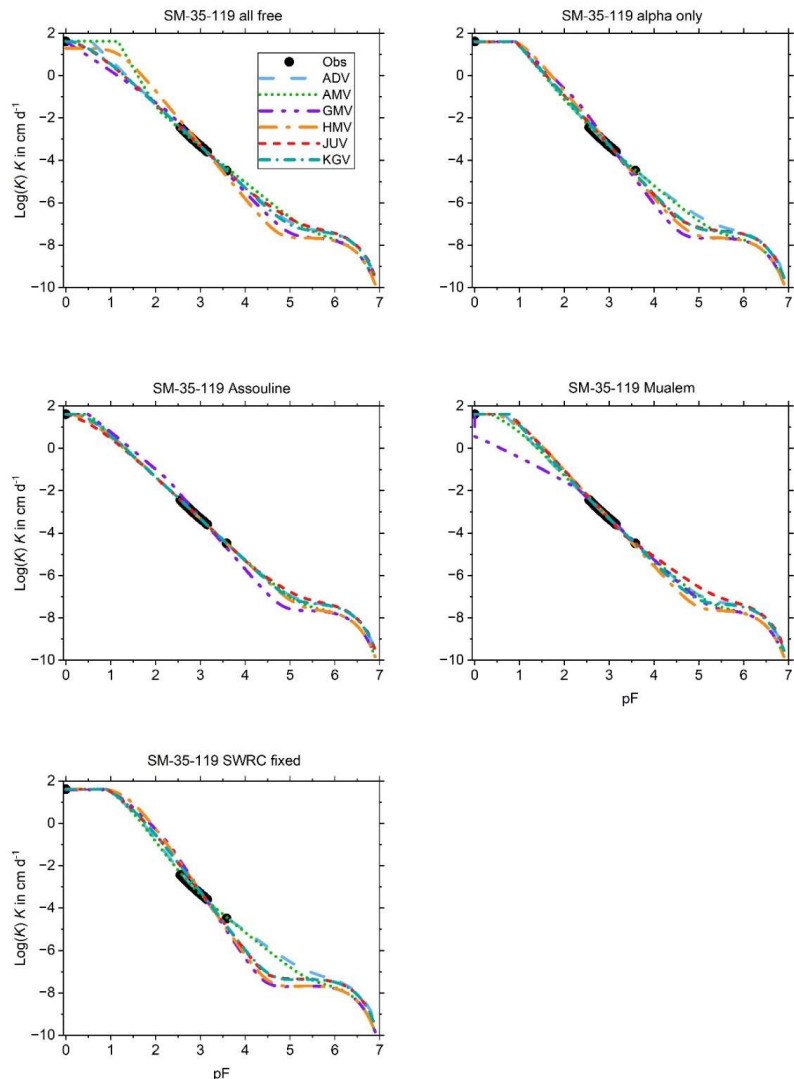

**Figure B7: As Fig. B2, but for SM–35–119.**





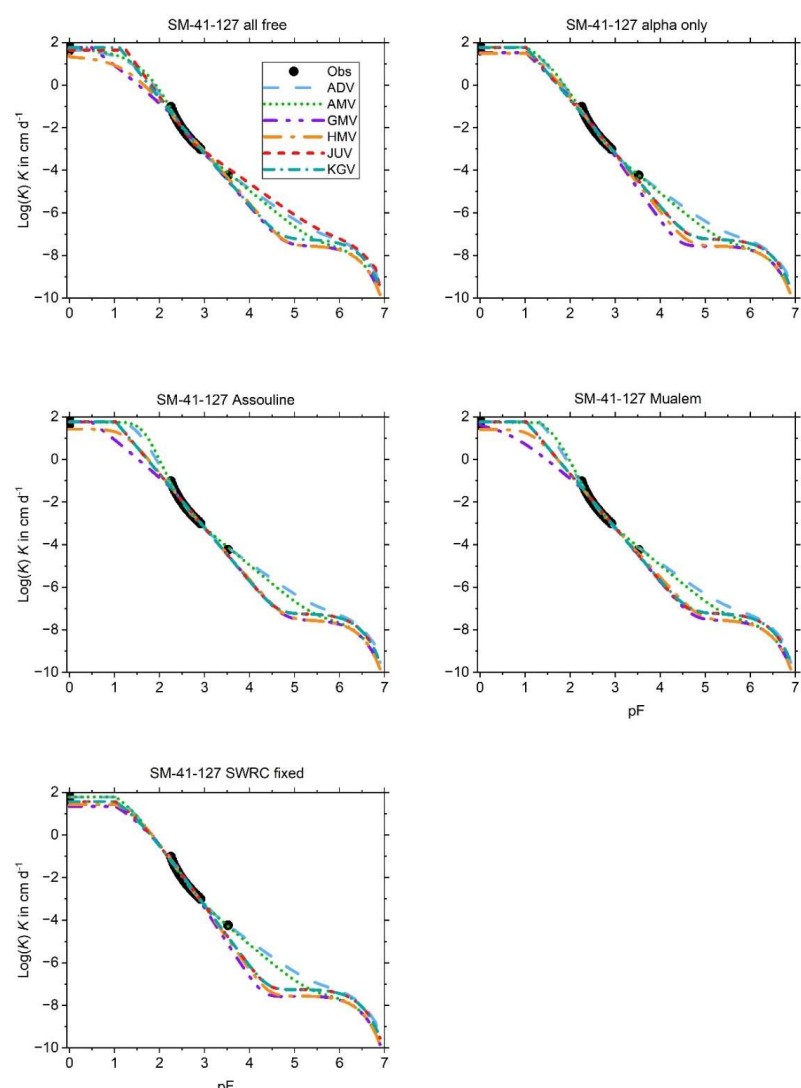


**Figure B8: As Fig. B2, but for SM–41–127.**



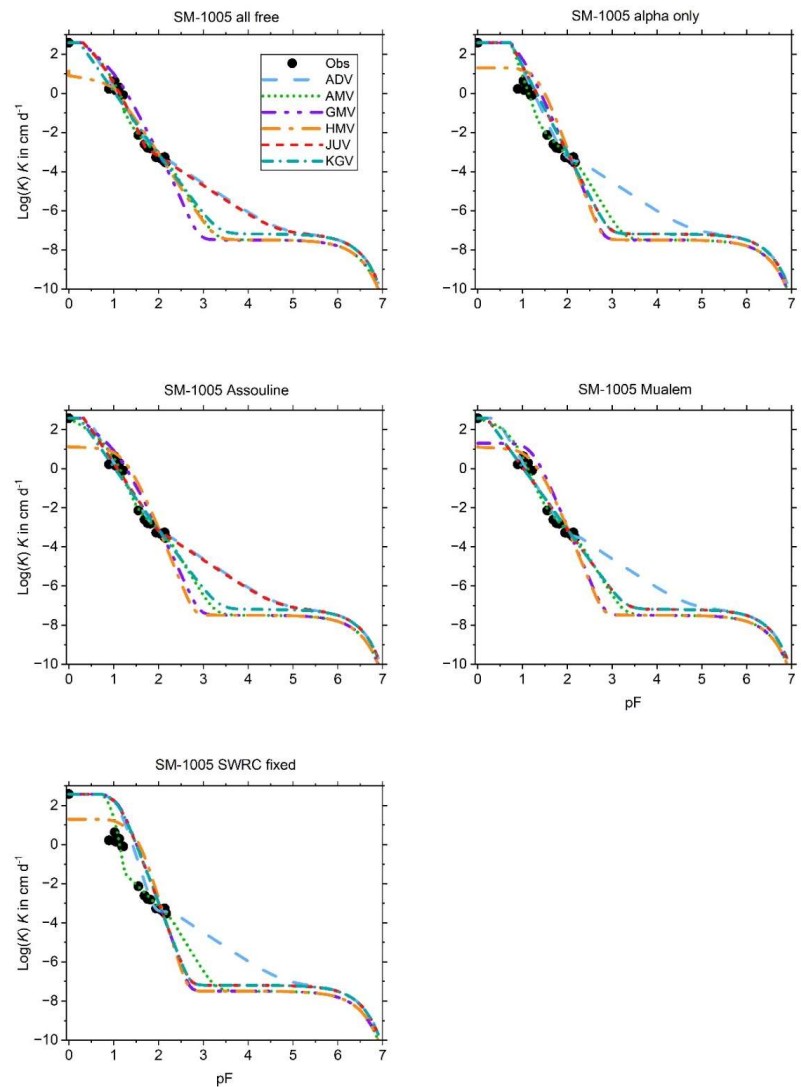

Figure B9: As Fig. B2, but for SM–1005.



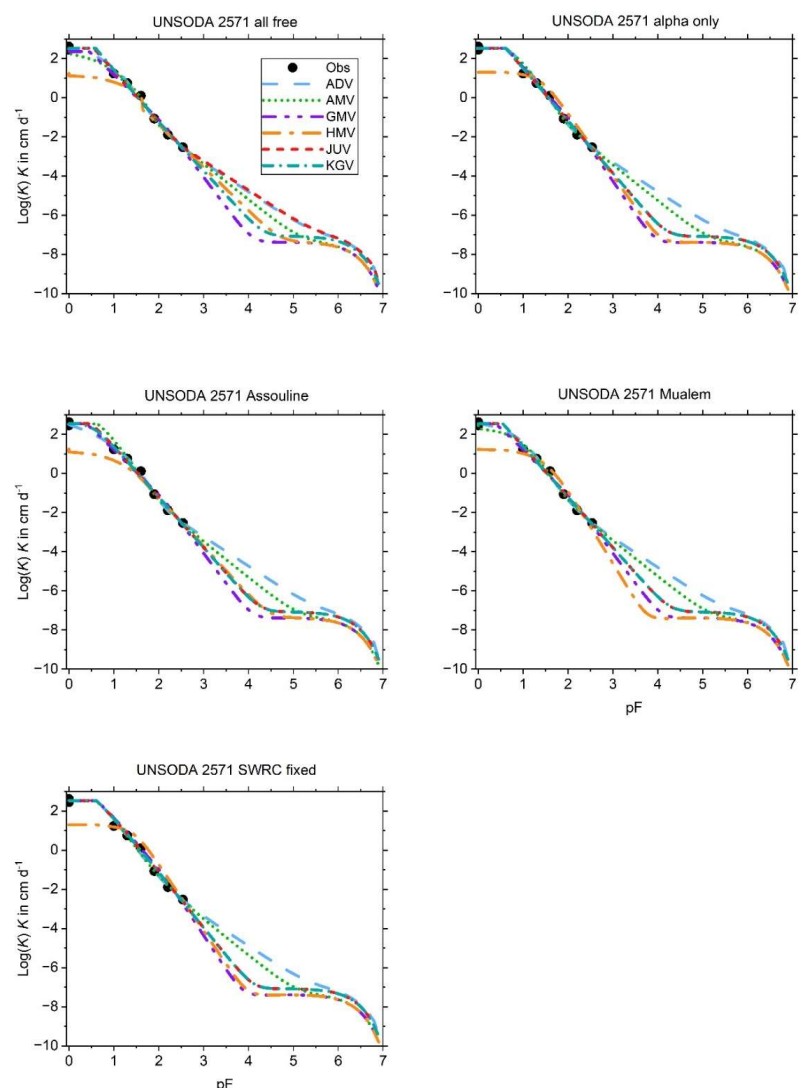

**Figure B10: As Fig. B2, but for UNSODA 2571.**



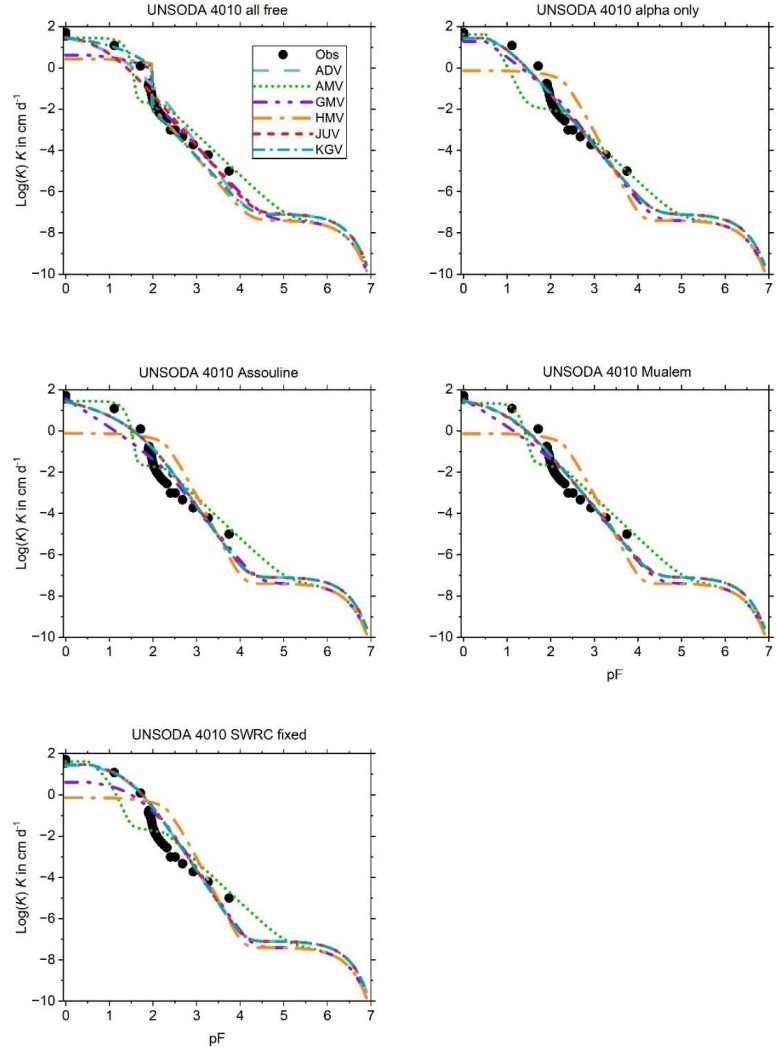

Figure B11: As Fig. B2, but for UNSODA 4010.



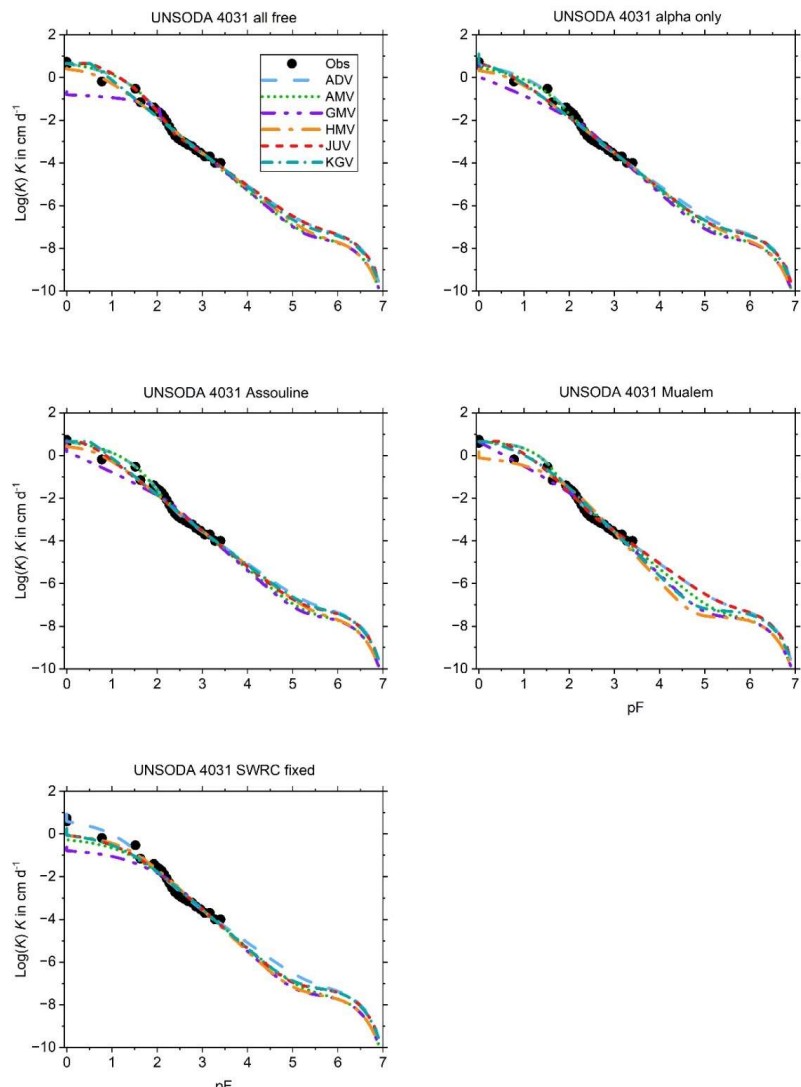

**Figure B12: As Fig. B2, but for UNSODA 4031.**



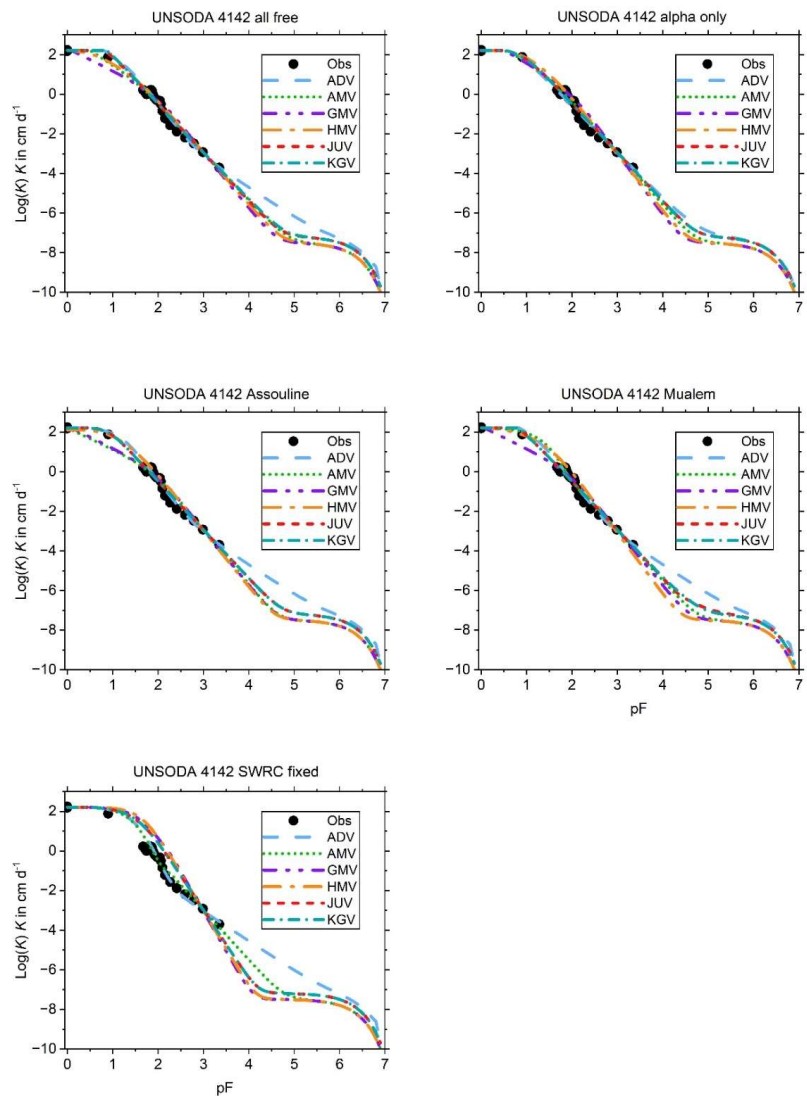

Figure B13: As Fig. B2, but for UNSODA 4142.



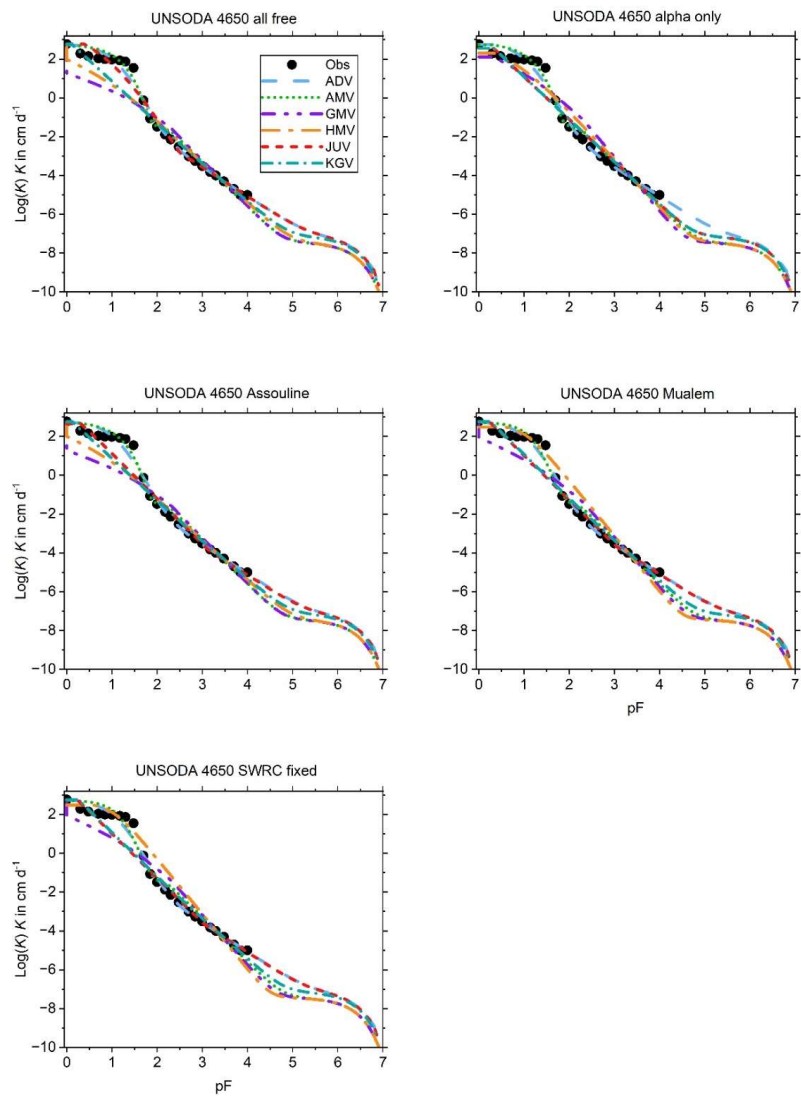

Figure B14: As Fig. B2, but for UNSODA 4650.



**Code and data availability**

The codes for RIAfitter and KRIAfitter as well as their User Manuals and example input and output files can be downloaded from de Rooij (2024c) and de Rooij (2024b), respectively. All input and output data as well as Excel files with processed data can be downloaded from de Rooij (2024d).

**Competing interests**

The author declares that he does not have any competing interests.

**Acknowledgements**

The author thanks Tobias Weber for making available the data of the test soils used in the paper.

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
