# Peer review of "Fitting the junction model and other parameterizations for the unsaturated soil hydraulic conductivity curve: KRIAfitter version 1.0"

_EGUsphere, 2024_

## Author Comment (AC1)

GMD paper on the junction model. Reviewer 1 (Katsutoshi Seki) plus reply

Below, the reviewer comments are in italics, and my replies in regular font.

**General Comment**

*This paper presents a range of models for unsaturated soil hydraulic conductivity curves (UHCCs), differing in how they combine approaches across three conceptual domains. Building on the author's previous work (de Rooij, 2024a), the paper introduces a new model, JUV, alongside earlier models (ADV, AMV, GMV, HMV, KGV), as summarized in lines 399–404. The use of the RIAfitter and KRIAfitter programs for model comparison is commendable, and it is particularly valuable that all data, code, and results are openly provided. The full description of the fitting procedures enhances the reproducibility of the research.*

*For each model, five parameter fitting strategies were tested, as outlined in lines 370–374. Among these, methods 2 and 5 rely on SWRC parameters that have already been fitted from retention curve data, whereas methods 1, 3, and 4 re-fit all parameters, including those of the SWRC. This is somewhat confusing, given that line 188 states: "Before KRIAfitter 1.0 can be run to determine the values of the parameters of the chosen UHCC model for a particular soil, the parameters of the SWRC of Eqs. (1a–d) need to be fitted using RIAfitter 2.0." If SWRC parameters are already known, it is unclear why they are re-fitted during UHCC fitting. If these parameters change during the UHCC fitting, then the SWRC curve would no longer match the originally fitted SWRC. This raises the question of whether the SWRC parameters in the SWRC equations are treated as independent from those in the UHCC equations (e.g., is the "n" in SWRC distinct from the "n" in UHCC?). From a modeling standpoint, it would seem more logical to fix the SWRC parameters and only fit the additional UHCC parameters, as is done in methods 2 and 5.*

Thank you for your thoughtful review.

This comment helped me identify the source of the confusion about the fitted SWRC and UHCC parameters. Before proposing a remedy, please allow me to offer some thoughts on the relevance of SWRC parameter values for the UHCC.

The predictive power of the fitted values for the SWRC for the values of the same parameters for the UHCC is often limited. As a general rule, I therefore do not recommend to assume that values of $\alpha$, $n$, and $h_{ae}$ fitted for the SWRC are necessarily valid for the UHCC as well. In principle, a code can be developed that allows one to fit all parameters on data from both curves, as RETC (van Genuchten's parameter fitting code) allowed for fitting van Genuchten's (1980) retention curve simultaneously with Mualem's (1976) conductivity curve. In all cases that I saw for which RETC was used with this option, the fits were not very good for either curve. I therefore did not implement this.

When using KRIAfitter, the user can choose the extent in which to use fitted values of SWRC parameters as fixed parameters for fitting the UHCC, as is illustrated by the various sets of fitting parameters, as the reviewer correctly notes.

Now, back to the point of the confusion about refitting parameters that were already fitted to the SWRC data points. To remove the source of confusion, I propose to give the parameters that appear in both the SWRC and the UHCC but can have different values different labels for the SWRC and the UHCC. The best way to so is probably adding the subscript $\theta$ to the

SWRC parameters, analogously to the distinction between $h_{j\theta}$ and $h_j$. I also propose to add the following to the start of Section 3 (Fitting the model parameters), together with additional minor edits in the text at the start of that section.

'The junction model (or any of the other UHCC model accommodated by KRIAfitter) does not require that the parameter values fitted for the SWRC are assumed to be valid for the UHCC as well. Nevertheless, physical consistency between the SWRC and the UHCC requires that $\theta_s$ and the matric potential at which liquid water is no longer present in the soil (calculated as $(1+c)h_d$ for the SWRC) are the same for both curves.'

The final sentence of the first paragraph of Section 3 can be revised as follows:

'Hence, $\theta_s$ is fixed at the SWRC value and $h_d$ is calculated according to Eq. (2b) from SWRC parameters that are provided on input.'

Equation 2b will heave to be inserted in the text below Eq. (2) in Section 2.2, which has to be relabeled (2a). That proposed revision is:

Rewrite start

The intrinsic hydraulic conductivity of water in films is modeled according to Peters (2013).

$$K_a(h) = K_{s,a} \cdot \begin{cases} 0, & h \leq h_d \\ \left(\frac{h}{h_a}\right)^{-1.5}, & h_d < h \leq h_a \\ 1, & h > h_a \end{cases} \tag{2a}$$

$K_{s,a}$ (LT$^{-1}$) is the value of $K_a$ when the domain with adsorbed water is completely filled, and $h_a$ (L) is the matric potential at which this occurs. The value of the exponent is adopted from Peters (2013). Note that $K_a(h)$ abruptly drops to zero at $h_d$ (the matric potential at oven-dryness, L), but $K_a$ at that matric potential is so small that this will generally be insignificant for practical use. The need to carry over the correction of $h_{d\theta}$ in Eq. (1a) results in the following equality.

$$h_d = (1 + c)h_{d\theta} \tag{2b}$$

Rewrite end

***Specific Comments***

1. *Table 1: Based on RMSE, the most flexible parameter set (method 1) yields the best results in nine cases, while the second-most flexible (method 3) does so in one case. Since optimization aims to minimize RMSE, and method 1 likely encompasses the parameter space of method 3, it is unclear why method 3 would ever outperform method 1. If method 1 includes all of method 3's parameters, then the optimal set*

*found by method 3 should also be attainable by method 1. This discrepancy could be due to method 1's parameter space not actually covering that of method 3, or because the fitting algorithm failed to locate the global optimum within the broader space of method 1.*

I noted this too. Prof. Seki is correct in stating that the parameter space of set 3 is wholly contained in that of set 1. In other fits, I also saw that constraining the parameter space can occasionally give better results. Having dimensions in the parameter space that do not improve the minimum appear to let the search algorithm run astray, and reduce its ability to locate the minimum, even though the complexes in parameter spaces with more dimensions contain more points.

The code returns the number of reflection, contraction, and random points required to achieve convergence, and from this it became clear to me that reduced parameter spaces considerably improve efficiency. This is consistent with the observation that a reduced parameter space (i.e., with at least one parameter fixed) can give better fits if the parameter values at the location of the global minimum coincide with the value of at least one fixed parameter for the fit in the reduced parameter space.

2. *As mentioned in the general comment, I believe that fixing all SWRC parameters during UHCC fitting is a reasonable approach. In this context, method 2 fixes the SWRC parameter α, but only method 5 fixes all SWRC parameters. I would be interested in seeing more variations on this fixed-parameter approach—such as versions assuming Assouline's τ = 0.0 or Mualem's γ = 2.0—alongside method 5. The author's model in fact resembles the model proposed by Peters (2013), which originally used γ = 2.0. The equation involving γ and τ is specifically intended to describe the capillary range. If the focus is solely on the capillary range, as in this study, it may be sufficient to fix γ = 2.0, without fitting both γ and τ. This could be tested as a simplification, though it is ultimately up to the author whether to include such comparisons.*

As I explained above, I think the quality of the fits of the UHCC will often improve if one does not fix the parameters that appear in the equations for both the SWRC and the UHCC are fixed to their values for the SWRC. Table 1 illustrates this by showing that that approach only gave the best fit (based on Akaike's Information Criterion) in a single case.

Mualem's model fixes both $\gamma$ and $\tau$. When the SWRC parameters are also fixed, this leaves only $K_{s,c}$ (and $K_{s,a}$ for some UHCC models) as fitting parameter(s). I did some trial fits with this, when experimenting with different sets of fitting parameters, and found that the results were generally rather poor, so I discarded that option as one of the sets to be used in the test of the code.

The five sets of fitting parameters all have a basis in the literature. GMD requires a test of the code in a model description paper such as this, which I provided through these five sets of fitting parameters.

I hope that people will use the model and the code to carry out studies as the one proposed by Prof. Seki, but this paper is a model description paper, not a paper in which the model is used for a particular application, which GMD does not publish. Therefore, exploring the validity of fixing a set of model parameters if one is only

interested in a specific range of the matric potential, as Prof. Seki proposes for the capillary range, is interesting, but beyond the scope of a model description paper in this journal.

The paper introduces a junction model with one branch for capillary flow, another for film flow, and a separate conductivity function for vapour flow. The work builds on a SWRC in which both capillary and adsorbed water are represented. I therefore do not understand the statement that this study focuses on the capillary range.

**Technical Comment**

1. *Equation 5a: Is the parameter "a" defined somewhere? Should this be "$D_a$"?*

Thank you for spotting this. The *D* dropped out when the Word file was converted to pdf before submitting the manuscript to the GMD website. This will be corrected (I hope to Word-to-pdf conversion will not corrupt it again).

**References**

- *Peters (2013): http://doi.org/10.1002/wrcr.20548*
- *Seki et al. (2023): https://doi.org/10.2478/johh-2022-0039*

The latter reference does not appear in the comment.

---

## Author Comment (AC2)

GMD manuscript on junction model. Report by Reviewer 2 and reply.

Below, the reviewer comments are in italics, and my replies in regular font.

*The manuscript presents a new junction model (JUV) to parameterize the unsaturated soil hydraulic conductivity curve (UHCC). The model assumes a switch from capillary flow to film flow at a critical matric head. In addition the manuscript documents a Fortran code, KRIAfitter, that estimates the parameters in the new JUV given measured values of hydraulic conductivity and matric head/water content. This code also includes several other UHCC models.*

*The idea of the junction model presents an interesting simplification for combining capillary flow and film flow. The provided fortran code for fitting of UHCC models ensures reproducibility and can be a useful tool for the community.*

*Presenting both, the new model and the code in one manuscript is challenging and unfortunately not successful. Some necessary information is missing or provided later in the manuscript. The manuscript needs to be reworked carefully to be able to review it. Below are some comments that may help improve the manuscript.*

The GMD website states the following about model development papers:

'The main paper should describe both the underlying scientific basis and purpose of the model and overview the numerical solutions employed. The scientific goal is reproducibility: ideally, the description should be sufficiently detailed to in principle allow for the re-implementation of the model by others, so all technical details which could substantially affect the numerical output should be described. Any non-peer-reviewed literature on which the publication rests should be either made available on a persistent public archive, with a unique identifier, or uploaded as supplementary information.'

I saw no other way of describing the underlying scientific basis (the model equations) and overview the parameter fitting code than in the way I did, providing both with adequate explanations. Below, I will outline how I believe the paper can be revised to improve the combined objective of presenting the junction model and the code to fit its parameters.

I appreciate the careful reading and the resulting comments. I give explanations where needed, and propose clarifications or rewritings to improve the paper.

*Specific comments:*

*Lines 86-87: "This correction needs to be applied to hd in the logarithmic branch of the SWRC as shown in Eq. (1a). No correction is needed in the UHCC." Equations 2 and 6 show c as well. So c is used in the UHCC. Please clarify.*

I see the reviewer's point. I meant to convey that, with the correction in the logarithmic branch SWRC, no additional corrections have to be made elsewhere. Nevertheless, the corrected logarithmic branch needs to be used everywhere, otherwise one can end up with some liquid water still present at a given matric potential according to the SWRC, with that

liquid water having a conductivity of zero according to the UHCC, for instance. I propose to rephrase as:

'This correction needs to be carried over to the expression for the soil hydraulic conductivity to liquid water in the UHCC.'

*Lines 98-109: I agree that "Simple addition of film and capillary conductivities may therefore not be accurate." However, the proposed simplification is clearly not accurate also. This should be discussed as well.*

In the literature of recent years, the additivity attribute of capillary and film conductivity is increasingly posited with a degree of confidence that suggests that this is an accurate representation of the physical reality, not a useful approximation. In de Rooij (2024a), I laid out a systematic critique of the assumption of the additivity attribute. In developing this critique, I found that it is fundamentally impossible to develop a physically-based model for the UHCC based on domain conductivities, as I reported in that paper. If one views the additivity of domain conductivities strictly as a tool for arriving at good overall fits of the UHCC instead of a physical representation of the nature of the soil hydraulic conductivity, then the junction model arises naturally as another tool, but with one parameter less. I can add the following explanation to the text:

'… not be accurate. De Rooij (2024a) showed that it is fundamentally impossible to develop a physically-based model for the UHCC based on domain conductivities. The junction model introduced here does not claim to have a more solid physical bases than other multi-domain UHCC models, but it is simpler than the existing models. To implement the junction model, the conductivity expressions were adapted as outlined below.'

*Equation 5a: What is a? Is this supposed to be Da? Please introduce Da then.*

Thank you for spotting this. The *D* dropped out when the Word file was converted to pdf before submitting. This will be corrected (I hope to Word-to-pdf conversion will not corrupt it again), and $D_a$ will be explained in the text below the equation.

*Equations 5b-d: units don't match*

These are all empirical equations that were fitted to approximate the temperature dependency of the three quantities involved. As is usual with this kind of equations, each term is implicitly multiplied by a constant equal to 1 that is assigned units that ensure dimensional consistency, but I have never seen a text where these constants and their units are explicitly declared. Because the values of the fitting constants obviously depend on the units of the dependent variable (the target variable on the left-hand-side of the equations) and the independent variable (the temperature in these cases), I declared the units of all involved variables in lines 141-145. This conforms to the established practice, so I prefer to keep the text as it is.

*Lines 167: is hd not fitted? Please clarify.*

The explanation is given in lines 193-196, and in a proposed edit of the start of Section 3. With the added explanation about the need to transfer the correction in $h_d$ from the SWRC to the UHCC in response to an earlier comment, I think this is now adequately clear in this part of the text, even before the later explanation.

*Lines 188-192: The split into two separate programs seems to limit the application. Could it make sense to fit alpha and n based on both the SWRC and UHCC?*

The need to fit the SWRC before the UHCC is on purpose. In principle, a code can be developed that allows one to fit all parameters on data from both curves. For instance, RETC (van Genuchten's parameter fitting code) has an option that allows for fitting van Genuchten's (1980) retention curve simultaneously with Mualem's (1976) conductivity curve. In all cases that I saw, the resulting fits were not very good for either curve. I therefore did not implement this.

The predictive power of the fitted values for the SWRC for the values of the same parameters for the UHCC is often limited. Having two codes gives the user maximal flexibility in choosing the extent in which to use fitted values of SWRC parameters as fixed parameters for fitting the UHCC. (In principle, the other way around is possible as well, but I am not aware of anyone attempting that, probably with good reason).

I fitted the RIA and the various models for the UHCC to many soils, and I found it easy to transfer the output (parameter values) of RIAfitter to the input of KRIAfitter, both as fixed parameter values of the UHCC and as independent SWRC parameter values that KRIAfitter uses to calculate the water content for a given matric potential). It is particularly convenient if one wishes to fit several UHCC models. Simply set up the input file for KRIAfitter for the first UHCC model, and run the code. Then, copy the input file, change the three-character code that identifies the desired conductivity model in the copy, and run KRIAfitter with the updated copy of the input file.

*Line 196: If alpha and n are already fitted as part of the SWRC, why are they not fixed? Or is this an iterative process?*

Reviewer 1 remarked on this too. To clear up the confusion, I propose to label the parameters that appear in both the SWRC and the UHCC but can have different values differently for the SWRC and the UHCC. The best way to so is probably adding the subscript $\theta$ to the SWRC parameters, analogously to the distinction between $h_{j\theta}$ and $h_j$.

On the need to be able to fit these parameters separately for the SWRC and the UHCC, see my response to the previous comment, my reply to Reviewer 1, and the brief explanation in lines 168-169 and 171-172. I do not recommend to assume that values of $\alpha$, $n$, and $h_{ae}$ fitted for the SWRC are necessarily valid for the UHCC as well. The user can test a fit based on this assumption by indicating in the input for KRIAfitter that the values of $\alpha$, $n$, and $h_{ae}$ given on input should not be changed, and inspecting the resulting fit. The values of any subset of these parameters can then be allowed to be fitted to see how much the fit improves. Note that $\theta_s$ is not fitted at all by KRIAfitter, and therefore automatically takes the SWRC value. The matric potential at which no liquid water is present anymore is set to $(1+c)h_d$, as calculated from the SWRC parameters. This is needed for physical consistency between the SWRC and the UHCC. In my reply to reviewer 1 I propose an improved explanation of this.

*Line 204-205: it is not clear what these parameters are.*

The fitting parameters of the junction model were declared earlier, in line 165-167. They are clearly identified as such, so I am not sure if I need to repeat that here. In line 169, the 7[th] fitting parameter for the more complicated multidomain models is mentioned. With this review in hand, I realize that that parameter needs a bit more explanation because it stems

from de Rooij (2024a). I therefore propose the following rewriting of the sentence in lines 169-170:

'The saturated conductivity of adsorbed water, $K_{s,a}$, is a fitting parameter for the multi-domain UHCC models presented by de Rooij (2024a). Because it is a derived parameter in the junction model (see Eq. (4)), the junction model has one parameter less than the models that average or add domain conductivities,'

*Line 226 and following: Why is the SCE algorithm needed to determine the optimal parameters. This is a global algorithm to avoid local minima. Why is it not possible to use a simple gradient based method?*

Reply: When I started this work, I was asked by fellow soil physicists at another institution why I considered coding a gradient-based model, now that global search models are available and computational power is no longer limiting. The reviewer takes the opposite view. Opinions on this matter apparently diverge, depending on how one weighs the arguments in favor of either approach. I chose a global search algorithm after consulting with colleagues of my institute that are well-versed in parameter optimization, and I am happy with the quality of the fits. I am not the only one to use SCE, by the way: see the SoilHyP package (https://rdrr.io/cran/SoilHyP/) of Dettman and coworkers.

*Line 232 (as an example): Very specific settings for the code like "but only if input variable FewComplexes is set to 'T' on input" should be limited to the user manual.*

The GMD guidelines quoted above state: '… all technical details which could substantially affect the numerical output should be described.' The settings I discuss in Section 3.3 are meant to affect the numerical output, and therefore belong in the paper, according to my reading of the guidelines.

Section 3.3 explains how the user can configure the fitting process, so I do not want to eliminate the connection with the input completely. Tom address this concern by the reviewer, I therefore propose to improve readability by removing the details of the settings of the input parameters from the text of Section 3.1 (for variable 'FewComplexes') and section 3.3 (for the other Boolean input), and instead collect that information in a table. The rewrite I have in mind will also make it clearer that Section 3.3 focuses on the set of binary choices that the user will have to make for each run of KRIAfitter.

This is the proposed revised version of Section 3.3:

Rewrite start

[revised manuscript text omitted]

Rewrite end

*Line 356: What pressure plate data is referred to here?*

I propose to rewrite this sentence to clarify this:

'… Because soil water retention data points obtained with pressure plate extractors (Dane and Hopmans, 2002) were found to be unreliable…'

Dane, J. H., and Hopmans, J. W.: Pressure plate extractor. In: Dane, J. H., and Topp, G. C. (editors), Methods of soil analysis. Part 4. Physical methods, Soil Scinece Society of America, Madison, Wisconsin, USA, pp. 688-690, doi: 10.2136/SSSAbookser5.4, 2002.

*Lines 370-374: Here the 5 different sets of parameters that were fitted are introduced. They are later referred to inconsistently. Sometimes as set 1-5, sometimes as 'Mualem' or 'alpha only'. Please be consistent and introduce the naming convention here.*

Good idea, thank you for the suggestion. I also propose to replace the list of sets by a table. The name of each set will then appear in one of the table columns.

*It is not clear what "Ks,a (when applicable means)". Ks,a is calculated using equation 4.*

I agree this needs clarification. See my response to the comment pertaining to lines 204-205to see how I propose to address this.

*In set 2: what about parameter n? Set 2 also lists hd to be used from the SWRC, the other lists don't mention it.*

In set 2, all critical values of the matric potential (air-entry value, junction point, oven-dryness) are set to the values fitted for the SWRC. That is why $h_d$ was included, although $h_d$ is fixed for the other sets as well. In the new table I propose below, this is handled differently is response to this comment.

By fixing both $h_j$ and $h_d$ in set 2, the value of $n$ is determined as well, through Eq. (1c). Of the SWRC parameters, only $\alpha$ remains as a fitting parameter (hence the name 'alpha only').

I propose to add the following explanation that text directly below the new table (included as well) that will replace the list of sets:

Rewrite start

| Table 2. The sets of fitting parameters used in the model evaluation. N.B. $\theta_s$ and $h_d$ are fixed for all sets. | | | |
|---|---|---|---|
| Set nr. | Set name | Fixed parameters | Fitting parameters |
| 1 | all free | none | $h_{ae}$, $\alpha$, $n$, $K_{s,c}$, $K_{s,a}$ (when applicable), $\gamma$, $\tau$ |
| 2 | alpha only | $h_{ae}$, $h_j$ as for the SWRC | $\alpha$, $K_{s,c}$, $K_{s,a}$ (when applicable), $\gamma$, $\tau$ |
| 3 | Assouline | $\tau = 0.0$ (Assouline, 2001) | $h_{ae}$, $\alpha$, $n$, $K_{s,c}$, $K_{s,a}$ (when applicable), $\gamma$ |
| 4 | Mualem | $\gamma = 2.0$, $\tau = 0.5$ (Mualem, 1976) | $h_{ae}$, $\alpha$, $n$, $K_{s,c}$, $K_{s,a}$ (when applicable) |
| 5 | SWRC fixed | $h_{ae}$, $\alpha$, and $n$ as for the SWRC | $K_{s,c}$, $K_{s,a}$ (when applicable), $\gamma$, $\tau$ |

In set 2, all critical values of the matric potential ($h_d$, $h_{ae}$, and $h_j$) are set to the values fitted for the SWRC. By fixing both $h_j$ and $h_d$, $n$ is determined as well, through Eq. (1c). Of the SWRC parameters, only $\alpha$ remains as a fitting parameter (hence 'alpha only').

Rewrite end

*Line 388: pF was not introduced*

In soil physics, pF is an established term that will be familiar to all that use the SWRC and the UHCC. But I will be happy to add its definition in the text:

'… pF (defined as $\log(-h)$ with $h$ in cm equivalent water column) of oven-dryness …'

*Lines388-390: This is unclear. What lower weight?*

I see your point. I propose the following clarification:

'… Because the retention data points in the dry range are unreliable (see above), their weighting factors are smaller than those of the other data points. With the pF (defined as $\log(-h)$ with $h$ in cm equivalent water column) of oven-dryness fixed at 6.8, this results in fitted dry branches that are drier than …'

*Lines 390-393: Where do these specific values for two soils come from?*

The text gives specific values for one soil only (Pachappa). As the text states, the value of $h_{ae}$ is the result of the parameter fitting process (by RIAfitter v 2.0), and the dataless range between $h = 0$ and $h = -47.0$ cm comes from the data points for Pachappa reported in the

literature. The first sentence of section 4.1 gives the reference to the source of the data sets, and the data points are shown in the graphs of Appendix B, as the text states. I do not know how to modify this text to clarify this further, all information seems to be there.

*Lines 399-404: Here the different methods the JUV is compared against are introduced. This should be done in the methods section. Additionally, this lacks crucial information. The methods all refer to (de Rooij, 2024a). Does this mean these are all recently introduced methods? The Kosugi model is established. What are the modifications? Please clarify. Also, why not compare to a simple established method?*

Yes, all conductivity models in de Rooij are new, even the Kosugi model, because de Rooij (2024a) augmented it with a vapor conductivity. Also, as was already explained in line 118, parameter $\kappa$ in Kosugi's model was set to one. De Rooij (2024a) has all information needed about every UHCC model other than the junction model. Therefore, no information is missing.

I believe that the reviewer perhaps confuses a model description paper, as described on the GMD web page, with a regular research paper. According to the GMD website, a model description paper does not have a Methods section, but ideally has a section with a test of the model the paper describes. The text to which this comment pertains is part of Section 4 of the paper, labeled 'Model evaluation'. The test described here is part of that evaluation, and is therefore in the correct position in the paper.

This paper contributes to an on-going debate in the literature about UHCC models. I started participating in this debate on the instigation of reviewers and one editor that worked on my papers on the RIA parameterization for the SWRC (the relevant papers were published in HESS and are quoted in the manuscript). I invite the reviewer to read the discussion in the pages of HESS to see why it is relevant to include in the evaluation a comparison with new models, represented here by the unweighted sum of the domain conductivities. I found in de Rooij (2024a) that the recently developed multidomain models of the UHCC had fundamental shortcomings, specifically the implicit assumption of parallel domains, the incorrect use of bulk conductivities, the impossibility of finding the correct weighting factors to be used in the averaging, and even the fundamental impossibility to find the correct way to average the domain conductivities. I developed the averaging models, both to highlight these shortcomings and to partially remedy them. The new, imperfect models I developed did not improve the fits compared to the unweighted additive model. With that in mind, it is only logical to include in the model evaluation the additive model as a representative for the modern conductivity models, and the Kosugi model as a representative of conventional models. One of the parameter sets that were fitted had its conductivity parameters fixed to those of Mualem's (1976) model, which is the most popular to date, and will be considered by almost all as 'a simple established method'. It therefore appears to me that the reviewers' suggestion is already accommodated in the paper.

*The following discussion of the results is difficult to follow. Please at least refer to the Figures with their Figure number.*

This section is meant to be read while studying the figures in Appendix B. It guides the reader to the most salient points as she or he studies the figures. I intentionally limited the text to the most important points of the graphs pertaining to each soil, because a more extensive discussion of the finer details adds little information, and is quite boring, to be frank. I will add the figure references. Thank you for that suggestion.

In addition, I propose to add the following addition at the start to clarify the nature of the analysis:

'The main points to observe from a qualitative analysis of the collection of graphs in Appendix B follow.'

*What is the evaluation if fits are good or not based on? Quantitative measures? Qualitative features? If so, which features?*

I assume the reviewer refers to the brief discussion of the fits for all soils that was also the subject of his previous comment. This (brief) segment of the text only focuses on the optics of the fits (qualitative features, in the terminology of the reviewer). The addition of the leading sentence proposed above clarifies this. I included this segment because the strict focus on the magnitude of goodness-of-fit criteria that follows this section does not explain why some fits are better than others.

When one reads the text while having the figures at hand, it becomes instantly clear which qualitative features are discussed. They vary from graph to graph, and are mainly determined by the range of the data points, the shape of the observed and fitted curves, and the way in which the conductivity near saturation is approximated (fitted based on a saturated conductivity that was either measured or estimated from SWRC properties, or fitted to unsaturated conductivity data only).

*Lines 454-455: This seems to be related to the range the observations cover. To me to does not make sense to include datasets that do not cover the dry range. Please clarify.*

I do not understand why the reviewer believes this. Rehovot sand has a distinct change of the slope in the observed UHCC (Fig. B4) that can only be reproduced if seven parameters are fitted.

As an aside, the body of literature that emerged in this century about more sophisticated UHCC models suffers from a lack of data in the dry range because the methodology to do so does not exist, or has a very poor accuracy. The soil physics community is aware of the problem, but has not yet solved it.

*Line 462: Here the Akaike's Information Criterion is introduced. It is later implied that this accounts for the number of parameters fitted. Please introduce earlier in the methods.*

As I explained above, a model description paper does not have a Methods section. KRIAfitter (and RIAfitter v 2.0) fit only one model at the time. Therefore, the objective function that minimizes the RMSE will also minimize the associated AIC. For the description of the fitting algorithm, the AIC is therefore irrelevant. In contrast, after multiple models have been fitted, AIC gains importance: one can select the best model based on the RMSE if one does not care about the number of fitting parameters, or based on the AIC if one worries about overparameterization. Therefore, I believe it is appropriate to introduce the AIC in the section where different UHCC models are compared.

*Line 480-484: Here the range of the underlying data is discussed. It could make sense to discuss the ranges also in terms of the calculated hj. This would allow the reader to better understand these ranges.*

This section describes the rationale for selecting the soils for which a more detailed analysis is carried out, based strictly on properties of the observed SWRC and UHCC, not on any properties of the fits. This was deliberate. When selecting cases for a more detailed analysis of the fitting results, I think it is fundamentally flawed to base the selection on one of these fitting results *a priori*. The risk for conscious or unconscious bias is real, and if you choose the fitting result used to select the cases, you may end up focusing on the aspects of the observed and fitted soil hydraulic properties that are of limited relevance for practical situations.

Apart from this fundamental objection, I do not see how taking in consideration the value of $h_j$ leads to an improved understanding of the range of the observations. I believe the presence or absence of a measured saturated conductivity matters much more. And because the bulk soil hydraulic conductivity tends to drop sharply in the wet range, when the largest, most conductive pores empty as the soil dries, data points near $h_{ae}$ are very important. The latter point is a major issue with the HyProp set-up for conductivity measurements, for instance.

*Figure 1: please add to the caption which method was used for each of the graphs.*

I can do that.